# FedVeer: Self-Adaptive Skew Estimation for Robust Federated Learning

Yun Xin [1] [#]   Bangqi Pan [1] [#]   Jianfeng Lu [2] [*]   Shuqin Cao [3]   Gang Li [4] [*]   Guanghui Wen [5]

## Abstract

Federated Learning (FL) enables collaborative model training across decentralized clients, but its performance often degrades under non-IID data distributions, particularly in the presence of data skew. Existing approaches mitigate this issue by estimating client skew via kernel density estimation over neighboring model updates, which preserves privacy and reduces communication costs. However, such approaches suffer from two fundamental limitations: bias toward skewed majority clients due to fixed neighborhood structures, and vulnerability to noise-induced perturbation in kernel space. To address these challenges, we propose FedVeer, a skew-aware FL framework based on self-adaptive kernel density estimation with $k$-free neighborhoods. FedVeer dynamically determines the neighborhood size via max-margin learning to mitigate majority-client bias, and further incorporates Kalman filtering to stabilize margin estimation under noisy updates, with a high-probability theoretical guarantee on margin deviation. Extensive experiments on real-world datasets demonstrate that FedVeer consistently outperforms four baselines, achieving up to 6.36% accuracy improvement and reducing noise-induced degradation by up to 6.01%.

## 1. Introduction

Federated learning (FL) has emerged as a paradigm for training a global model across decentralized clients while

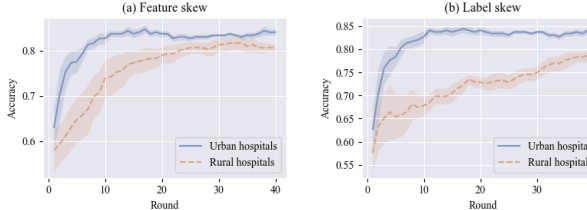

*Figure 1.* Accuracy of FedAvg on the UCI Heart Disease (Dua & Graff, 2017) under (a) feature skew and (b) label skew. Urban hospitals perform large-scale random sampling, whereas rural hospitals sample from a feature-restricted subset under feature skew, or from a biased proportion of positive and negative classes under label skew.

preserving data privacy by keeping data local (McMahan et al., 2017). Despite its promise, FL often suffers from significant performance degradation when client data are non-identically distributed (non-IID), a phenomenon commonly induced by location-level constraints and client-specific preferences across diverse dats sources (Reisizadeh et al., 2020; Huang et al., 2023). In particular, data skew, such as imbalanced feature distributions or biased label proportions across clients, poses a fundamental challenge to model convergence and generalization. For example, in healthcare applications, urban hospitals typically collect larger and more diverse datasets than rural hospitals, resulting in highly skewed data distributions across institutions. Such imbalance can substantially slow convergence and degrade the final model performance, as empirically illustrated in Figure 1 (Vo et al., 2022; Wang et al., 2025).

To mitigate data imbalance under non-IID distributions, it is essential to quantify the degree of data skew for each client. Some studies require clients to upload feature prototypes or local label statistics, which increases privacy risks and incurs substantial communication and computation overhead in decentralized FL settings (Wang et al., 2025; Zhang et al., 2022). To alleviate these concerns, kernel-based methods have been proposed to evaluate client skew directly from their model updates, without accessing raw data or local statistics (Li et al., 2021). However, these approaches rely on fixed kernel parameters, implicitly assuming that severely skewed clients constitute only a small minority (Zhang et al., 2024). When skew is widespread, such fixed neighborhood structures cause kernel-based estimates to be dominated

[1]School of Computer Science and Technology, Wuhan University of Science and Technology, China [2]Hubei Province Key Laboratory of Intelligent Information Processing and Real-time Industrial System, Wuhan University of Science and Technology, China [3]Key Laboratory of Social Computing and Cognitive Intelligence (Dalian University of Technology), Ministry of Education, China [4]College of Computer Science, Inner Mongolia University, China [5]School of Automation, Southeast University, China. Correspondence to: Jianfeng Lu <lujianfeng@wust.edu.cn>, Gang Li <gli@imu.edu.cn>.

*Proceedings of the 43 $^{rd}$ International Conference on Machine Learning*, Seoul, South Korea. PMLR 306, 2026. Copyright 2026 by the author(s).

by majority-skewed clients, leading to biased aggregation. Moreover, skew evaluation is inherently vulnerable to noise arising from stochastic optimization and partial participation, which introduces perturbations regardless of whether skew estimation is based on raw samples, local statistics, or kernel representations (Huang et al., 2023; Vo et al., 2022). Consequently, designing a noise robust aggregation framework that mitigates data skew without accessing local data remains a critical and increasingly urgent challenge.

Taking these limitations into account, an adaptive kernel-based approach for skew-aware FL is required, yet it faces two fundamental challenges. First, *bias toward skewed majority clients in kernel-based skew estimation.* Since kernel similarity is often used to quantify clients informativeness or consistency, estimates can be dominated by skewed majority clients, causing clients with more balanced data distributions to be underrepresented and even misidentified as anomalous (Zhang et al., 2024). Second, *kernel-based evaluation is inherently sensitive to noise.* Because kernel computation relies on similarity estimation, stochastic noise can distort the geometry of model updates in kernel space, leading to biased or unstable similarity measures (Wang et al., 2020). Together, these challenges make it particularly difficult to design an adaptive kernel mechanism for skew-aware FL when majority clients are both imbalanced and noisy.

To overcome these challenges, we propose FedVeer, a skew-aware Federated learning framework via self-adaptiVe kernel density estimation. Concretely, FedVeer comprises a self-adaptive kernel density estimation module for skew-aware aggregation, a Kalman filter-based margin stabilization mechanism for noise robustness, and a theoretical framework guarantees margin stability under noisy updates. Unlike prior kernel-based approaches with fixed neighborhoods, FedVeer dynamically adjusts neighborhood structures to mitigate majority-client bias and incorporates noise-robust margin learning to stabilize kernel-based skew estimation. Our main contributions are summarized as follows:

- We develop a noise-free variant of FedVeer that self-adaptively determines the optimal neighborhood size $k$ via max-margin learning and uses it to estimate kernel density for client aggregation weights. Building on the noise-free formulation, we further extend FedVeer to a noise-robust variant by incorporating a Kalman filter with an optimal Kalman gain into the neighborhood size determination process, effectively mitigating noise-induced perturbations.

- We provide rigorous theoretical analysis for FedVeer. In particular, we prove the uniqueness of the solution to the max-margin learning problem. Moreover, we derive an upper bound on the probability that noise induces a significant deviation of the expected mar-

gin from its noise-free counterpart, and show that this bound is strictly tightened by Kalman filtering.

- We conduct extensive experiments on real datasets to validate the effectiveness and robustness of FedVeer. The results demonstrate that FedVeer's performance is strongly governed by the neighborhood size, achieving up to a $6.36\%$ accuracy improvement over four state-of-the-art baselines under severe data skew, while reducing noise-induced degradation in max-margin learning by up to $6.01\%$.

**Conflict of Interest Disclosure**    The authors declare that they have no known competing financial interests or personal relationships that could have appeared to influence the work reported in this paper.

## 2. Related Work

### 2.1. FL with Non-IID Data

FL operates under non-IID data distributions, which poses significant challenges to model training. In particular, the heterogeneity of local data leads to high diverse local updates, thereby impeding the convergence and performance improvement of the global model. Non-IID data distributions can lead to data skew, which includes feature skew and label skew, especially when clients originate from different data sources, where the skew becomes more pronounced. Most existing methods of tackling data skew primarily focus on a single type of skew. For example, FedRDN addressed feature skew by leveraging feature distribution information (Yan et al., 2025), FedLC mitigated label skew through label distribution statistics (Zhang et al., 2022), while FedHEAL focused on fairness-aware aggregation mechanisms to handle domain skew (Chen et al., 2024). The major limitation of these methods is that they are tailored to specific types of data skew, which leads to unsatisfactory generalization performance when multiple skews coexist. In our work, we propose a skew-aware FL framework that does not rely on any probabilistic model of feature, labels, or their conditional distribution. Moreover, it does not require estimating feature or label distribution, enabling generalization in scenarios where multiple types of data skew coexist.

### 2.2. Mitigating Data Skew in FL

Most existing methods of tackling data skew primarily rely on client local feature or label statistics, which requires implicitly sharing sensitive client-side information. For example, FPL relied on class-related feature representations (Huang et al., 2023), FDSE exchanged aligned feature representations (Wang et al., 2025), and FedLC utilized local label distribution statistics as shared information to mitigate skew (Zhang et al., 2022). The major limitation of these

methods is their reliance on exposing client local variant information, which may lead to increased privacy risks and additional communication/computation overhead. In contrast to existing skew-aware FL methods that rely on client local feature or label statistics, we propose a kernel-based approach that evaluates the degree of client skew through local model parameter.

### 2.3. Kernel-based Skew Detection in FL

Kernel-based data detection methods are widely used in model analysis to characterize parameter distributions and detect heterogeneity among model parameters. For example, LoMar exploded kernel density estimation for outlier detection (Li et al., 2021), while CausalRFF designed a customized kernel to quantify causal effects (Vo et al., 2022). However, existing works typically adopt fixed kernel parameters, implicitly assuming that the skewed clients represent only a small fraction of the population. To address more general skew scenarios in which the majority of clients exhibit data skew, KFNN dynamically adjusted kernel parameters to adapt to the overall skew conditions across clients (Zhang et al., 2024). However, KFNN operates in a centralized setting and relies on direct access to global feature and label information. The design is incompatible with FL, which follows a decentralized paradigm where neither raw feature nor labels are accessible, and even sharing feature or label based statistics may introduce privacy risks. In our work, we propose an adaptive kernel for skew-aware FL that quantifies client-level and source-level heterogeneity in local model parameters, with kernel parameters dynamically adjusted according to the severity of data skew.

## 3. System Model

### 3.1. Preliminary

A standard FL setup consists of a set of clients $\mathcal{N} = \{1, \cdots, N\}$ and a central server that aggregates the model parameters collected from the clients. To capture data heterogeneity, suppose we have $S$ sources of data. These sources reside in different locations, and their distributions can be completely different (Vo et al., 2022). Each client $n \in \mathcal{N}$ has a private dataset $\mathcal{D}_n^s$ with $D_n^s$ training samples, originating from a data source $s \in \mathcal{S}$, as shown in Figure 2 step ①. For each data source $s$, we denote its dataset as $\mathcal{D}^s = \{(\mathbf{x}_i, y_i)\}_{i=1}^{c_s}$, where $c_s$ is the number of instances in source $s$, $x_i$ represents the $i$-th noisy feature vector of the $i$-th instance in $D^s$, and $y_i$ denotes the corresponding noisy label. Each client's local dataset is a subset of a $D^s$, and the $n_s$ clients associated with the same data source collectively constitute $D^s$.

For each round $t$, FL aims to collaboratively train a shared global model $\omega^t$. After receiving the global model parameter

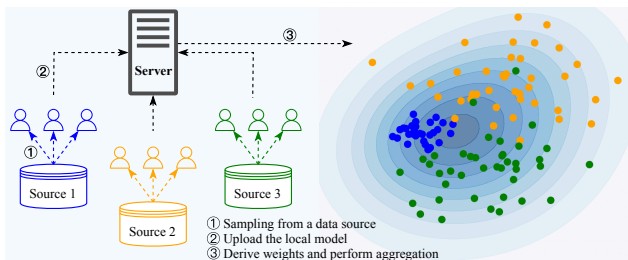

*Figure 2.* An overview of kernel-based aggregation for FL.

$\omega^t$, client $n$ performs local updates to obtain the updated model $\omega_n^{t+1}$:

$$\omega_n^{t+1} = \omega_n^t - \eta \nabla l(\omega^t, \mathcal{D}_n^s), \qquad (1)$$

where $\eta$ is the learning rate, and $\nabla l(\omega^t, \mathcal{D}_n^s)$ denotes the gradient of the local average loss $l(\omega^t, \mathcal{D}_n^s)$. After clients upload their local models, as shown in Figure 2 step ②, the server performs model aggregation as follows:

$$\omega^{t+1} = \sum \theta_n \omega_n^{t+1}, \qquad (2)$$

where $\theta_n$ is the aggregation weighted assigned to client $n$, reflecting its relative impact o the global model.

Two natureal choices for setting $\theta_n$ are the uniform weight $\frac{1}{N}$ and the proportional weight $\frac{D_n^s}{\sum_{n \in \mathcal{N}} D_n^s}$ (Li et al., 2024). However, these settings are unsuitable when some clients are abnormal or exhibit skewed data distributions, as they fail to down-weight updates from skewed client.

### 3.2. Problem Formulation

Recent studies have shown that kernel density estimation can be leveraged to quantify client data skew in FL, thereby guiding aggregation by assigning larger weights to more informative clients, as illustrated in step ③ of Figure 2. Specifically, for a client $n$, its kernel density is estimated from a neighborhood model set

$$\mathcal{W}_n^k = \{\omega_n^i\}_{i=1}^k, \qquad (3)$$

where $\omega_n^i$ denotes the $i$-th nearest neighbor of $\omega_n$, and $k$ is the neighborhood size. Existing kernel-based federated aggregation methods typically adopt a fixed neighborhood size $k$ when estimating kernel density. However, this design is suboptimal in heterogeneous federated environments. For data sources with a large number of clients, a larger $k$ tends to include more clients with similar local models, which may amplify majority-client bias when the underlying data distribution is skewed. Consequently, selecting an appropriate neighborhood size becomes critical for accurately estimating kernel density and mitigating skew during aggregation. We therefore consider an idealized noise-free

setting, where the optimal neighborhood size is determined by minimizing the expected prediction loss:

$$k^* = \arg\min_k \mathbb{E}[l(\hat{y}_i(k), y_i)],$$
$$\text{s.t. } \hat{y}_i(k) = h(\{\rho(\mathcal{W}_n^k)\}_{n=1}^N), \tag{4}$$

where $h(\cdot)$ denotes a hypothesis function, and $\hat{y}_i$ is the predicted label corresponding to $x_i$.

In practical federated learning, kernel density estimation may be corrupted by noisy client updates, which perturb the underlying similarity structure among models. Since the server cannot directly access local data or labels, it must ensure that kernel density estimation remains stable under noise. To this end, we impose a high-probability constraint that bounds the deviation between noisy and noise-free kernel density estimates:

$$\mathbb{P}(|\rho(\mathcal{W}_n^{k'}) - \rho(\mathcal{W}_n^k)| \geq \varepsilon) \leq \delta, \tag{5}$$

where $\mathcal{W}_n^{k'}$ and $\mathcal{W}_n^k$ denote the neighborhood sets under noisy and noise-free settings, respectively, $\varepsilon$ controls the estimation deviation, and $\delta$ controls the confidence level.

Combining adaptive neighborhood selection with robustness to noise, we formulate the following optimization problem:

$$\mathbb{P}_1: \ k^* = \arg\min_k \mathbb{E}[l(\hat{y}_i(k), y_i)],$$
$$\text{s.t. } \begin{cases} \hat{y}_i(k) = h(\{\rho(\mathcal{W}_n^k)\}_{n=1}^N), \\ \mathbb{P}(|\rho(\mathcal{W}_n^{k'}) - \rho(\mathcal{W}_n^k)| \geq \varepsilon) \leq \delta. \end{cases} \tag{6}$$

**Remark:** The proposed formulation enables kernel density estimation to adaptively quantify client contributions while ensuring robustness against noise-induced perturbations. By controlling the deviation probability, the formulation guarantees that aggregation weights are not excessively distorted under noisy client updates.

## 4. Noise-Free FedVeer

In this section, we introduce the core formulation of FedVeer under a noise-free setting, which serves as the foundation of our approach. We first characterize kernel density estimation for federated aggregation with a given neighborhood size, and then derive the optimal neighborhood size through max-margin learning.

### 4.1. Kernel-Based Weighting in FL

To evaluate the relative positions of client updates $\omega_i$ and $\omega_j$, we measure their distance using $l_2$-norm, i.e., $d(\omega_i, \omega_j) = \| \omega_i - \omega_j \|_2$. Then, we obtain the neighborhood model set for client $n$ by sorting all other clients in ascending order of their distances to $\omega_n$, denoted as $\mathcal{W}_n^k = \{\omega_n^i\}_{i=1}^k$, where $\omega_n^i$ as the model with the $i$-th smallest distances to $\omega_n$.

The kernel density estimation function can be used to estimate the local distribution of client $n$'s update $\omega_n$ based on its $k$-nearest neighborhood $\mathcal{W}_n^k$. We denote the estimated distribution of client $n$'s local model $\omega_n$ by $q_n$, and the estimation of $q_n$ is formally given as follows:

$$q_n = \frac{1}{k} \sum_{i=1}^k \mathcal{K}(\frac{d(\omega_n, \omega_n^i)}{h}), \tag{7}$$

where $\mathcal{K}(\cdot)$ is the kernel function, and $h$ denotes its bandwidth parameter. In particular, we choose $\mathcal{K}(\cdot)$ as a Gaussian kernel defined as follows:

$$\mathcal{K}(\frac{d(\omega_n, \omega_n^i)}{h}) = \frac{1}{\sqrt{2\pi h}} \exp(-\frac{d(\omega_n, \omega_n^i)}{2h}). \tag{8}$$

Intuitively, if a client update $\omega_n$ lies in a low-density area, its estimated density $q_n$ tends to be small; conversely, updates located in high-density yield larger $q_n$. However, this is not always guaranteed. If only a few neighbors fall within a high-density region, an update located in that region may still obtain a low $q_n$. If the number of neighbors in a low-density region is high, an update in that region can unexpectedly yield a relatively large $q_n$.

Based on the estimated a client-level density $q_n$, we further derive a source-level density $q^s$, which captures the average density of clients from data source $s$, defined as follows:

$$q^s = \frac{1}{N^s} \sum_{n \in \mathcal{N}_s} \frac{1}{k} \sum_{i=1}^k \mathcal{K}(\frac{d(\omega_n, \omega_n^i)}{h}). \tag{9}$$

To mitigate the negative impact of data skew clients on global aggregation, we assign each client $n$ a weight by jointly incorporating its client-level kernel density and the corresponding source-level kernel density:

$$\theta_n = \frac{q_n \sum_{s=1}^S q^s \mathbf{1}[n \in s]}{\sum_{n=1}^N q_n \sum_{s=1}^S q^s \mathbf{1}[n \in s]}, \tag{10}$$

where $\mathbf{1}[n \in S]$ is an indicator function that equals 1 if client $n$ is associated with source $s$, and 0 otherwise. These weighting rule assigns lower weights to clients whose local model updates lie near the distribution boundary within a data source, as characterized by low client-level kernel density. Moreover, clients from severely skewed data sources are further down-weighed, as these sources exhibit low source-level kernel density.

### 4.2. Self-Adaptive Neighborhood Size in FL

Let $\mathcal{Q}_k = \{q^s(k) | s \in \mathcal{S}\}$ denote the set of source-level kernel densities across all data sources. The adaptive kernel is selected by maximizing the margin $m(\cdot)$ between the

---

**Algorithm 1** Noise-Free FedVeer

---

1: **Input:** $\{\omega_1, \cdots, \omega_N\}$, $m = 0$, $s^* = 0$
2: **Output:** $k^*$
3: **for** $k = 1$ **to** $N$ **do**
4:     **for** $s = 1$ **to** $S$ **do**
5:        $q^s(k) \leftarrow \frac{1}{N^s} \sum_{n \in \mathcal{N}_s} \frac{1}{k} \sum_{i=1}^{k} \mathcal{K}(\frac{d(\omega_n, \omega_n^i)}{h})$
6:     **end for**
7:     **if** $s^* \neq 0$ and $\arg \max \mathcal{Q}_k \neq s^*$ **then**
8:        Terminate the loop
9:     **end if**
10:    **if** $\max \mathcal{Q}_k - sec \max \mathcal{Q}_k > m$ **then**
11:       $m \leftarrow \max \mathcal{Q}_k - \sec \max \mathcal{Q}_k$
12:       $k^* \leftarrow k$, $s^* \leftarrow \arg \max \mathcal{Q}_k$
13:    **end if**
14: **end for**

---

largest and the second-largest kernel densities:

$$\mathbb{P}_2: k = \arg \max_k m_k, \tag{11}$$
$$\text{s.t. } m_k = \max \mathcal{Q}_k - \sec \max \mathcal{Q}_k.$$

Different data source exhibit different degrees of distribution alignment. Let $s^*$ denote the data source with best distribution alignment.

**Theorem 4.1.** *Under assumption A.1 as defined in Appendix A.1, there exists a unique solution to problem $\mathbb{P}_2$.*

After establishing the existence and uniqueness of the solution to problem (11), we propose a noise-free FedVeer algorithm for self-adaptive neighborhood size selection in FL. In Algorithm 1, we traverse different neighborhood size $k$ to compute the source-level density scores for all data source (lines 4-8). According to Assumption A.1, the data source with the maximum source-level density score corresponds to the best-aligned data source in the initial stage, i.e., $k < \hat{k}$. Then, the iterate process terminates once the best-aligned data source changes (lines 9-11). Obtain $k^*$ from $\mathcal{C}$ according to Eq. (11) and proceed to the next iteration over $k$ (lines 12-16).

## 5. Noise-Robust FedVeer

In this section, we consider FedVeer under a noisy setting, where local client updates may perturb the max-margin neighborhood learning process. We first establish a high-probability upper bound on the deviation of the learned margin from its noise-free expectation induced by noisy updates. Building on this analysis, we introduce a Kalman filter–based margin stabilization mechanism to further tighten the bound and improve robustness.

### 5.1. Margin Deviation Bounds under Noisy Updates

We analyze the impact of noise on the margin when clients' local updates are affected by noisy data. Let $\tilde{m}_k$ denote the margin under noisy data and $m_k$ denote the margin under clean data. The error induced by noise is defined as $v_k = \tilde{m}_k - m_k$, with variance denoted by $\beta$. There are several challenges in proceeding with this analyze: First, kernel density estimation provides only an empirical approximation of the local update distribution, rather than the true underlying data-generating distribution. Second, noisy affecting local updates is inherently stochastic, making the resulting perturbation on the margin computation random. To address these challenges, we derive an asymptotic threshold $\xi$ to characterize the difference between the margin under clean data and that under noisy data, which quantifies the margin bias under worse-case noise condition. Consequently, we apply McDiarmid's inequality (Long et al., 2024; McDiarmid, 1989) to derive a probabilistic upper bound on the deviation of the margin from its clean expected value, with the impact of noisy local updates captured by the bounded difference $\xi$.

**Theorem 5.1.** *A local update changes the margin by at most $\xi$, i.e., $|m_k - \tilde{m}_k| \leq \xi$. Then, for any $\gamma > 0$,*

$$\mathbb{P}(|m_k - \mathbb{E}[m_k]| \geq \gamma) \leq \exp(-\frac{2\gamma^2}{N\xi}), \tag{12}$$

*where $\xi_n = \frac{(k + N_s - 1)[1 - \exp(-\frac{\beta^2}{h})]}{N^s (2\pi)^{r/2}}$.*

**Remark:** The above theorem indicates that the deviation bounds depended on three factors: (i) A larger neighborhood sizes $k$ amplifies the variance of margin estimation under noisy updates, increasing instability. (ii) A larger number of clients within a data source $N^s$ stabilizes margin by providing more abundant data. (iii) A larger observation error variance $\beta$ increases the likelihood of large margin deviation.

### 5.2. Reducing Margin Deviation via Kalman Filtering

In this section, we use a Kalman filter to mitigate the effect of noise on the margin. Let $\hat{m}_k$ denote the estimated margin after filtering. Since the estimation error arises from multiple random perturbations and exhibits no directional drift, it can be reasonably regarded as an unbiased random process, i.e., $\mathbb{E}[e_k] = 0$. Based on the Central Limit Theorem, the aggregation of these perturbations leads the error distribution to approximate a Gaussian shape. Moreover, modeling the noise as an unbiased Gaussian process is consistent with the Kalman Filters' linear-Gaussian noise assumption. Therefore, we can model the estimation error $e_k = \hat{m}_k - m_k$ as $\mathcal{N}(0, p_k)$. Besides, let $\alpha$ as the process noise variance that quantifies evaluated margin unreliability.

To evaluate the current margin estimate $\hat{m}_k$, we first initialize the prior estimate for this round, denoted as $\hat{m}_k^-$, i.e., $\hat{m}_k^- = \hat{m}_{k-1}$. Then, as the real margin $m_k$ may drift randomly across rounds, the uncertainty of the estimate $\hat{m}_k$ should be increase to $\alpha$. A larger $\alpha$ implies a more uncertain and less reliable estimate $\hat{m}_k$, leading the filter to rely more heavily on the new observation $\tilde{m}_k$. Note that the margin estimate may exhibit a state jump when the data source switches between consecutive rounds. In this case, we amplify the prior uncertainty $p_k$ using a forgetting factor $\lambda$, which reduces the confidence in historical predictions and encourages the filter to rely more on the current observation. That is, we initialize the prior estimate error variance $p_k^-$ as:

$$p_k^- = \{1 + \mathbf{1}[s_k \neq s_{k-1}](\frac{1}{\lambda} - 1)\}p_{k-1} + \alpha. \quad (13)$$

Next, we derive the Kalman gain $\kappa_k$, which trades off the relative influence of the predicted margin and the observation, and updates the estimate along their discrepancy with a magnitude determined by $\kappa_k$.

$$\hat{m}_k = \hat{m}_k^- + \kappa_k(\tilde{m}_k - \hat{m}_k^-). \quad (14)$$

We next derive the Kalman gain $\kappa_k$ that minimizes the estimation error, along with the corresponding update of the error variance $p_k$.

**Lemma 5.2.** *The Kalman gain that minimize the mean-squared error $\mathbb{E}[(\hat{m}_k - m_k)^2]$ is given by*

$$\kappa_k = \frac{p_k^-}{p_k^- + \beta}. \quad (15)$$

*With the optimal gain, the estimation error variance is update as*

$$p_k = (1 - \kappa_k)^2 p_k^- + \kappa_k^2 \beta. \quad (16)$$

Then, we propose a noise-robust variant of FedVeer that mitigates margin noise by incorporating a Kalman filter. As shown in Algorithm 2, for each $k$, we first initialize $\mathcal{Q}_k$ and $\tilde{m}_k$ (line 3-9), and then apply Kalman filtering to stabilize the margin estimation (line 10-14). The filtering process consists of two stages for each $k$: In the estimate stage, a prior margin estimate is obtained based on the historical information (line 10-11). In the update stage, this estimate if refined by combining the prior estimate with the noisy observation through the Kalman gain (line 12-14), yielding a minimum mean-square-error estimate of the margin. Based on the filtered estimate $\hat{m}_k$, we further derive a probabilistic upper bound on the deviation of the margin from its clean expected value under noisy updates, as stated below.

**Theorem 5.3.** *A local update changes the margin after filtering by at mose $\zeta$, i.e., $|m_k - \hat{m}_k| \leq \zeta$. Then, for any $\gamma > 0$,*

$$\mathbb{P}(|m_k - \mathbb{E}[m_k]| \geq \gamma) \leq \exp(-\frac{2\gamma^2}{N\zeta}), \quad (17)$$

---

**Algorithm 2** Noise-Robust FedVeer

1: **Input:** $\alpha$, $\beta$, $N$, $p_0$, $\hat{m}_0$, $\mathcal{M} = \emptyset$, $\{\omega_1, \cdots, \omega_N\}$
2: **Output:** $k^*$
3: **for** $k = 1$ **to** $N$ **do**
4:     $\mathcal{Q}_k \leftarrow \emptyset$
5:     **for** $s = 1$ **to** $S$ **do**
6:         $q^s(k) \leftarrow \frac{1}{N^s} \sum_{n \in \mathcal{N}_s} \frac{1}{k} \sum_{i=1}^k \mathcal{K}(\frac{d(\omega_n, \omega_n^i)}{h})$
7:         $\mathcal{Q}_k \leftarrow \mathcal{Q}_k \cup q^s(k)$
8:     **end for**
9:     $\tilde{m}_k \leftarrow \max \mathcal{Q}_k - \sec \max \mathcal{Q}_k$
10:    $\hat{m}_k^- \leftarrow \hat{m}_{k-1}$
11:    $p_k^- = \{1 + \mathbf{1}[s_k \neq s_{k-1}](\frac{1}{\lambda} - 1)\}p_{k-1} + \alpha$
12:    $\kappa_k \leftarrow \frac{p_k^-}{p_k^- + \beta}$
13:    $\hat{m}_k \leftarrow \hat{m}_k^- + \kappa_k(\tilde{m}_k - \hat{m}_k^-)$
14:    $p_k \leftarrow (1 - \kappa_k)^2 p_k^- + \kappa_k^2 \beta$
15:    $\mathcal{M} \leftarrow \mathcal{M} \cup \{\hat{m}_k\}$
16: **end for**
17: $k^* \leftarrow \max \mathcal{M}$

---

*where $\zeta = \sqrt{\frac{\beta p_k^-}{p_k^- + \beta}}$, and $\zeta < \xi$*

**Remark:** After filtering, the deviation bounds is tightened by replacing $\xi$ with $\zeta$, which captures the reduced uncertainty induced by Kalman filtering. This improvement stems from the optimal combination of prior estimates and noisy observations, leading to a tighter probabilistic bound on margin deviation.

## 6. Experiments

### 6.1. Experimental Setup

**Datasets.** We estimate FedVeer on four datasets including MNIST (LeCun et al., 2021), Fashion-MNIST (Xiao et al., 2017), FEMNIST (Caldas et al., 2018), and CIFAR-10 (Krizhevsky, 2009) to verify its effectiveness under data skew. We further validate FedVeer on 34 industry-oriented datasets from the CEKA toolkit (Zhang et al., 2015), spanning domains such as manufacturing, biomedicine, finance, and healthcare, to demonstrate its generalization capability across diverse real-world data skew. Detailed experimental settings for datasets are provided in the Appendix E.1.

**Baselines.** To show the effectiveness of FedVeer, we compare it with four baselines, including FedAvg (McMahan et al., 2017), LoMar (Li et al., 2021), FedLC (Zhang et al., 2022), and FedRDN (Yan et al., 2025), under scenarios where the majority of clients exhibit data skew. FedAvg serves as a standard baseline without addressing data skew. LoMar mitigates data skew through fixed-size neighborhood kernel density estimation under the assumption that only a minority of clients exhibit skewed data. FedLC mitigates

*Table 1.* Accuracy comparison across different neighborhood sizes.

| Dataset | $k = 5$ | $k = 8$ | $k = 10$ | $k = 13$ | $k = 15$ |
|---|---|---|---|---|---|
| anneal | 0.929 | **0.961** | 0.949 | 0.920 | 0.946 |
| audiology | 0.633 | **0.742** | 0.545 | 0.686 | 0.717 |
| autos | 0.634 | **0.780** | 0.683 | 0.756 | **0.780** |
| balance-scale | 0.896 | 0.875 | 0.887 | **0.899** | 0.896 |
| biodeg | 0.872 | 0.877 | **0.888** | 0.873 | 0.880 |
| breast-cancer | 0.690 | 0.703 | 0.690 | **0.741** | 0.693 |
| breast-w | 0.974 | 0.971 | **0.976** | 0.964 | 0.964 |
| car | **0.908** | 0.887 | 0.908 | 0.859 | 0.865 |
| credit-a | 0.900 | 0.898 | 0.903 | 0.888 | **0.904** |
| credit-g | 0.756 | **0.776** | 0.745 | 0.769 | 0.733 |
| diabetes | 0.787 | 0.790 | **0.806** | 0.799 | 0.781 |
| heart-c | 0.803 | 0.804 | **0.845** | 0.834 | 0.811 |
| heart-h | 0.795 | **0.847** | 0.818 | 0.811 | 0.806 |
| heart-statlog | 0.851 | 0.788 | 0.833 | **0.865** | 0.778 |
| hepatitis | 0.871 | **0.936** | 0.892 | 0.871 | 0.871 |
| horse-colic | 0.774 | 0.802 | 0.816 | 0.800 | **0.833** |
| hypothyroid | **0.966** | 0.955 | 0.950 | 0.958 | 0.959 |
| ionosphere | 0.926 | 0.944 | **0.956** | 0.930 | 0.949 |
| iris | 0.940 | 0.936 | 0.933 | 0.933 | **0.967** |
| kr-vs-kp | 0.957 | **0.962** | 0.938 | 0.941 | 0.952 |
| labor | 0.694 | 0.722 | **0.872** | 0.667 | 0.861 |
| letter | 0.807 | 0.808 | 0.807 | **0.809** | 0.808 |
| lymph | **0.858** | 0.842 | 0.829 | 0.782 | 0.807 |
| mushroom | 0.992 | 0.993 | 0.991 | 0.993 | **0.995** |
| segment | 0.918 | **0.936** | 0.923 | 0.931 | 0.910 |
| sick | **0.955** | 0.953 | **0.955** | 0.954 | 0.954 |
| sonar | 0.914 | **0.939** | 0.857 | 0.873 | 0.897 |
| spambase | 0.933 | 0.935 | 0.937 | 0.936 | **0.943** |
| tic-tac-toe | 0.980 | **0.984** | 0.965 | 0.966 | 0.974 |
| vehicle | **0.722** | 0.672 | 0.671 | 0.653 | 0.642 |
| vote | 0.908 | **0.929** | 0.887 | 0.897 | 0.887 |
| vowel | 0.584 | **0.651** | 0.577 | 0.531 | 0.528 |
| waveform | 0.841 | 0.839 | **0.851** | 0.836 | 0.833 |
| zoo | 0.857 | **0.943** | 0.857 | 0.924 | 0.952 |

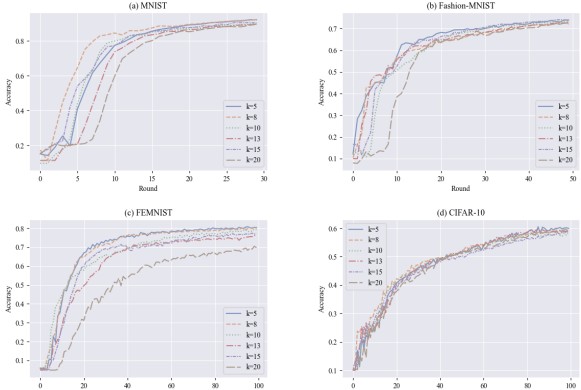

*Figure 3.* Accuracy comparison across neighborhood sizes on (a) MNIST, (b) Fashion-MNIST, (c) FEMNIST, and (d) CIFAR-10.

label distribution skew without addressing feature skew, whereas FedRDN mitigates feature distribution skew without addressing label distribution skew. Detailed description of the baselines are provided in the Appendix E.2.

### 6.2. Performance across Different Neighborhood Sizes

To verify the effect of neighborhood size on kernel density estimate and show the necessity of adaptive kernel estimation that dynamically adjusts the neighborhood size based on the margin, we compare the global accuracy at neighborhood sizes of 5, 8, 10, 13, and 15. This comparable neighborhood sizes are reasonable, as non-skew clients require higher kernel density scores than other clients to dominate their weight, thereby improving global model performance under data skew. To achieve this, the neighborhood size used for kernel density score estimation should be close to the number of non-skew clients.

In Figure 2 and Table 1, we observe that different neighborhood sizes lead to varying convergence speed and result.

Moreover, an optimal neighborhood size exists that yields either the highest. This is because margin maximization causes high-density client updates to dominate aggregation, so the neighborhood size should be tuned to the geometry of the update space rather than client count, and is not necessarily equal to the number of non-skew clients. When non-skewed clients produce highly consistent updates in the model update space, a small neighborhood size suffices to capture high-density neighbors and maximize the margin gap against skewed updates. When non-skewed client updates are more dispersed, larger $k$ can capture enough neighbors for stable density estimation, while these updates still form higher-density regions than skewed ones.

### 6.3. Performance Comparison with Baselines

In Figure 3 and Table 2, we compare accuracy against 5 baselines. As shown in Figure 3, our proposed framework FedVeer achieves higher accuracy and faster convergence. In Figure 3, FedLC (Zhang et al., 2022), FedRDN (Yan et al., 2025), and FedLSA (Fu et al., 2025) exhibit relatively poor performance because they are primarily designed to address specific forms of data heterogeneity and rely on local representation regularization or local statistical alignment. Consequently, they can only partially alleviate data skew and struggle when the majority of clients exhibit severe skewed distributions. In addition, LoMar (Li et al., 2021) and TPFL (Gu et al., 2026) employ kernel-density-based mechanisms with fixed kernel parameters, which are sensitive to dominant skewed update patterns and may overemphasize skewed clients during aggregation. In contrast, FedVeer achieves superior performance under severe skew by adaptively adjusting the kernel parameter according to the density structure of client model updates.

Table 2 shows that FedVeer consistently achieves the highest accuracy among CEKA platform datasets. We observe that our framework achieves superior performance on datasets

*Table 2.* Accuracy comparison across different baselines.

| Dataset | LoMar | FedLC | FedRDN | TPFL | FedLSA | FedVeer |
|---|---|---|---|---|---|---|
| anneal | 0.955 | 0.941 | 0.909 | 0.944 | 0.950 | **0.961** |
| audiology | 0.626 | 0.738 | 0.641 | 0.696 | 0.692 | **0.772** |
| autos | 0.637 | 0.390 | 0.608 | 0.684 | 0.585 | **0.724** |
| balance-scale | 0.896 | 0.896 | 0.896 | 0.889 | 0.896 | **0.899** |
| biodeg | 0.787 | 0.748 | 0.793 | 0.766 | 0.721 | **0.824** |
| breast-cancer | 0.522 | 0.670 | 0.693 | 0.719 | 0.551 | **0.756** |
| breast-w | 0.944 | 0.956 | 0.964 | 0.970 | 0.969 | **0.971** |
| car | 0.880 | 0.803 | 0.832 | 0.869 | 0.835 | **0.890** |
| credit-a | 0.823 | 0.798 | 0.811 | 0.847 | 0.843 | **0.872** |
| credit-g | 0.748 | 0.750 | 0.712 | 0.748 | 0.746 | **0.753** |
| diabetes | 0.750 | 0.772 | 0.742 | 0.788 | 0.763 | **0.790** |
| heart-c | 0.803 | 0.836 | 0.832 | 0.773 | 0.769 | **0.834** |
| heart-h | 0.769 | 0.806 | 0.703 | 0.810 | 0.788 | **0.814** |
| heart-statlog | 0.772 | 0.803 | 0.776 | 0.775 | 0.790 | **0.848** |
| hepatitis | 0.912 | 0.870 | 0.912 | 0.892 | 0.813 | **0.932** |
| horse-colic | 0.775 | 0.741 | 0.790 | 0.815 | 0.721 | **0.844** |
| hypothyroid | 0.941 | 0.941 | 0.937 | 0.940 | 0.936 | **0.949** |
| ionosphere | 0.669 | 0.733 | 0.723 | 0.808 | 0.723 | **0.824** |
| iris | 0.743 | 0.666 | 0.613 | 0.667 | 0.717 | **0.770** |
| kr-vs-kp | 0.919 | 0.929 | 0.912 | 0.912 | 0.941 | **0.948** |
| labor | 0.875 | 0.870 | 0.812 | 0.877 | 0.783 | **0.917** |
| letter | 0.857 | 0.845 | 0.856 | 0.855 | 0.857 | **0.859** |
| lymph | 0.835 | 0.856 | 0.795 | 0.848 | 0.840 | **0.860** |
| mushroom | 0.988 | 0.990 | 0.991 | 0.989 | 0.987 | **0.995** |
| segment | 0.900 | 0.881 | 0.893 | 0.888 | 0.891 | **0.901** |
| sick | 0.946 | 0.947 | 0.950 | 0.951 | 0.950 | **0.951** |
| sonar | 0.880 | 0.900 | 0.823 | 0.860 | 0.869 | **0.919** |
| spambase | 0.918 | 0.912 | 0.914 | 0.923 | 0.901 | **0.932** |
| tic-tac-toe | 0.819 | 0.832 | 0.851 | 0.834 | 0.838 | **0.862** |
| vehicle | 0.682 | 0.681 | 0.672 | 0.656 | 0.672 | **0.689** |
| vote | 0.919 | 0.919 | 0.908 | 0.906 | 0.917 | **0.924** |
| vowel | 0.907 | 0.899 | 0.869 | 0.904 | 0.899 | **0.911** |
| waveform | 0.845 | 0.847 | 0.696 | 0.850 | 0.840 | **0.851** |
| zoo | 0.857 | 0.904 | 0.780 | 0.904 | 0.903 | **0.904** |
| average | 0.826 | 0.825 | 0.791 | 0.840 | 0.820 | **0.866** |

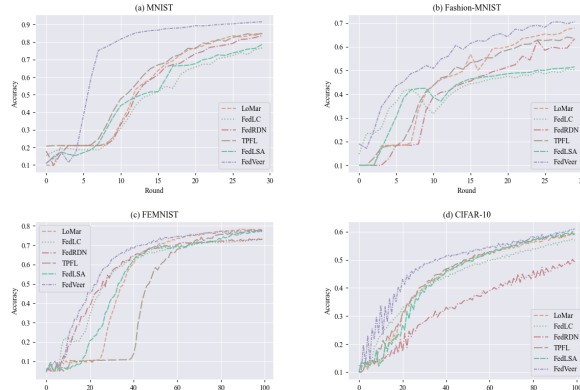

*Figure 4.* Accuracy comparison across different baselines on (a) MNIST, (b) Fashion-MNIST, (c) FEMNIST, and (d) CIFAR-10.

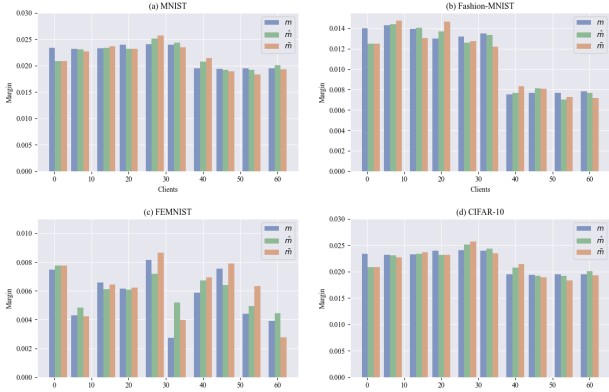

*Figure 5.* Margin under noisy defense on (a) MNIST, (b) Fashion-MNIST, (c) FEMNIST, and (d) CIFAR-10.

with more complex feature and label structures. In contrast, when the data structure is relatively simple, client data exhibit less pronounced skew, leading to accuracy comparable to that of baseline methods. This is because more complex data structures exhibit larger density variations, causing different neighborhood sizes to have greater impact on density estimation and accuracy.

### 6.4. Performance for Noisy Defense

To verify that the Kalman filter effectively reduces the impact of noise on the margin, we compare margin results across three settings: models without noise, models with noise before filtering, and models with noise after filtering. Because clients from different sources yield different kernel density scores, and if density structure within each source is relatively stable, then a larger client population in a source leads to higher density and thus higher kernel density score. Therefore, to amplify the effectiveness of the Kalman filter

across different sources, we increase the number of non-skew clients to 20, while keeping the total number at 60. Note that the majority of clients still exhibit data skew.

In Figure 3, margin values with noise after Kalman filtering are substantially closer to the noise-free margins than to the unfiltered noisy margins. Clients from the same source exhibit similar margins, which is evident on MNIST, Fashion-MNIST, and CIFAR-10. For example, the last 20 are non-skewed, their margins show lower variance and are nearly consistent. This phenomenon arises because the Kalman filter recursively evaluate predicted margins with noisy observed margins using an adaptive Kalman gain, producing a minimum mean-square error optimal estimate that smooths noise-induced fluctuations and let estimates closer to the noise-free values. Since different sources exhibit vibrating margins, we introduce a forgetting factor in the Kalman prediction step to reduce outdated noise accumulation and enhance recent-state tracking, enabling the predicted evaluation to better follow the current source's margin.

## 6.5. Ablation Study

FedVeer consists of a self-adaptive kernel density estimation module for skew-aware aggregation and a Kalman filter-based margin stabilization mechanism for noise robustness. We perform end-to-end ablation studies on four datasets under four settings: (i) without both modules, (ii) with only the Kalman filter-based margin stabilization mechanism, (iii) with only the self-adaptive kernel density estimation module, and (iv) with both modules enabled.

*Table 3.* Ablation study of different model components.

| Setting | MNIST | Fashion-MNIST | FEMNIST | CIFAR-10 |
|---------|-------|---------------|---------|----------|
| (i) | 0.909 | 0.699 | 0.377 | 0.567 |
| (ii) | 0.914 | 0.707 | 0.463 | 0.568 |
| (iii) | 0.921 | 0.731 | 0.608 | 0.588 |
| **(iv)** | **0.922** | **0.741** | **0.636** | **0.591** |

The results show that removing either component consistently degrades performance, indicating that the two modules provide complementary benefits. In particular, removing the adaptive kernel module causes substantial accuracy drops on heavily skewed datasets, showing that fixed neighborhood structures fail to capture the density geometry of client updates and are more easily dominated by skewed majority clients. Removing Kalman filtering also reduces performance, as noisy local updates perturb margin estimation and destabilize kernel-based aggregation. By adaptively learning neighborhood sizes and stabilizing noisy margins, the complete FedVeer framework achieves more reliable skew estimation and robust aggregation across heterogeneous federated environments.

## 6.6. Hyper-parameters Learning

**Kernel-related hyper-parameters learning.** We consider two kernel-related hyper-parameters $(h, k)$, where $h$ denotes the Gaussian kernel bandwidth and $k$ denotes the neighborhood size. Empirically, we evaluate $h \in \{x, x/2, x/4\}$, where $x$ denotes the median pairwise distance between client model updates. The results show that, for each dataset, the performance exhibits a consistent trend with respect to $k$ across different choices of $h$. This is because FedVeer relies on margin-based relative density comparison together with adaptive neighborhood selection, which partially mitigates the influence of $h$, while $k$ plays a more dominant role in shaping the density geometry of client updates.

**Kalman-related hyper-parameters learning.** We consider three Kalman-related hyper-parameters $(\lambda, \alpha, \beta)$, corresponding to the forgetting factor, process noise variance, and observation noise variance, respectively. To evaluate robustness, we vary $\lambda$, $\alpha$, and $\beta$ on four datasets and measure the error reduction ratio $\Delta = \frac{|m-\tilde{m}|-|m-\hat{m}|}{|m-\tilde{m}|}$, where $m$, $\tilde{m}$, and $\hat{m}$ denote the clean, noisy, and filtered margins, re-

*Table 4.* Kernel-related hyper-parameter learning.

| $h$ | $k$ | MNIST | Fashion-MNIST | FEMNIST | CIFAR-10 |
|-----|-----|-------|---------------|---------|----------|
| | 5 | 0.817 | **0.739** | **0.806** | **0.511** |
| $x$ | 8 | **0.865** | 0.731 | 0.787 | 0.508 |
| | 13 | 0.798 | 0.739 | 0.778 | 0.502 |
| | 5 | 0.813 | **0.738** | **0.634** | **0.398** |
| $x/2$ | 8 | **0.862** | 0.732 | 0.629 | 0.395 |
| | 13 | 0.794 | 0.738 | 0.604 | 0.378 |
| | 5 | 0.803 | **0.735** | **0.322** | **0.238** |
| $x/4$ | 8 | **0.850** | 0.725 | 0.307 | 0.225 |
| | 13 | 0.770 | 0.734 | 0.307 | 0.191 |

spectively. A larger $\Delta$ indicates stronger noise suppression and more effective margin stabilization. The results show that Kalman-based filtering consistently improves margin estimation across all settings. Increasing $\lambda$ generally improves $\Delta$, suggesting that stronger retention of historical margin estimates benefits stability. Smaller $\alpha$ values also achieve better performance, as overly responsive updates may amplify estimation fluctuations. In contrast, larger $\beta$ values improve $\Delta$ by assigning lower confidence to noisy margin observations, resulting in stronger smoothing and more stable margin estimation.

*Table 5.* Kalman-related hyper-parameter learning ($\times 10^{-3}$).

| $(\lambda, \alpha, \beta)$ | MNIST | Fashion-MNIST | FEMNIST | CIFAR-10 |
|----------------------------|-------|---------------|---------|----------|
| (**0.3**, 0.01, 0.02) | 0.071 | 0.064 | 0.538 | 0.010 |
| (**0.5**, 0.01, 0.02) | 0.119 | 0.107 | 0.898 | 0.016 |
| (**0.7**, 0.01, 0.02) | 0.161 | 0.144 | 1.211 | 0.022 |
| (**0.9**, 0.01, 0.02) | 0.195 | 0.175 | 1.465 | 0.027 |
| (0.7, **0.001**, 0.02) | 0.351 | 0.313 | 2.650 | 0.047 |
| (0.7, **0.005**, 0.02) | 0.220 | 0.197 | 1.656 | 0.030 |
| (0.7, **0.010**, 0.02) | 0.161 | 0.144 | 1.211 | 0.022 |
| (0.7, **0.020**, 0.02) | 0.110 | 0.099 | 0.826 | 0.015 |
| (0.7, 0.01, **0.01**) | 0.110 | 0.099 | 0.826 | 0.015 |
| (0.7, 0.01, **0.02**) | 0.161 | 0.144 | 1.211 | 0.022 |
| (0.7, 0.01, **0.05**) | 0.240 | 0.215 | 1.806 | 0.033 |
| (0.7, 0.01, **0.10**) | 0.300 | 0.268 | 2.259 | 0.040 |

## 7. Conclusion

In this paper, we have proposed FedVeer, which has introduced a self-adaptive neighborhood size $k$ for kernel density estimation to mitigate data skew. Specifically, we have adapted the neighborhood size in a self-adaptive manner via max-margin learning and assigns aggregation weights accordingly, favoring clients with more balanced data. To further enhance robustness, a Kalman filter is incorporated into the max-margin learning process to mitigate the impact of noisy data. After filtering, the margin deviation between clean and noisy has been upper bounded with higher probability than without filtering. Finally, extensive experiments on real datasets demonstrate that FedVeer improves accuracy and effectively mitigates the impact of noise.

## Acknowledgments

This work was supported in part by the National Natural Science Foundation of China (No. 62372343, 62325304, 62402352, U22B2046, U2541220, 62072411), in part by the Basic Research Program of Jiangsu (No. BK20253020), in part by the Jiangsu Provincial Scientific Research Center of Applied Mathematics (No. BK20233002), in part by the Open Fund of Key Laboratory of Social Computing and Cognitive Intelligence (Dalian University of Technology), Ministry of Education (No. SCCI2024TB02), and in part by the Open Project Funding of the Key Laboratory of Intelligent Sensing System and Security (Hubei University), Ministry of Education (No. KLISSS202406).

## Impact Statement

This paper presents a methodological contribution to FL, focusing on improving robustness under data heterogeneity and noisy client updates. The proposed approach may help enhance the reliability of federated learning systems in practical settings. We do not anticipate any significant negative societal impact arising directly from this work.

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

# A. Technical Preliminaries and Proof for Theorem 4.1

## A.1. Assumption on Source-Level Kernel Density Estimation

**Assumption A.1.** There exists a threshold $\hat{k}$ such that, for all $k < \hat{k}$ and for all $s \in \mathcal{S}$,

$$q^{s*}(k) \geq q^s(k), \tag{18}$$

where $s*$ denotes the cluster with the most balanced data distribution.

**Remark.** This assumption is commonly used in anomaly detection and accuracy evaluation in FL, and has also been widely adopted for sample similarity analysis in crowdsourcing (Luo and Wu, 2022; Li et al., 2021; Cao et al., 2022). It is reasonable to assume that clients with better distribution alignment attain higher kernel density scores for a given sample sizes, since high-accuracy clients tend to generate model updates with similar directions, whereas low-accuracy clients exhibit more diverse and less consistent update directions. The case where clients with low accuracy produce incorrect updates that still align in the same direction is a low-probability event and is thus omitted here.

## A.2. Monotonicity of Source-Level Kernel Density Estimation

**Lemma A.2.** *For each cluster $s \in \mathcal{S}$, $q^s(k)$ is monotonically non-increasing w.r.t. $k$, i.e.,*

$$q^s(k+1) \leq q^s(k). \tag{19}$$

**Proof.** We prove the result in two steps: first at the client level for the density $q_n(k), n \in \mathcal{N}$, and then extend it to the source level density $q^s(k), s \in \mathcal{S}$.

First, we analyze client-level monotonicity of density $q_n(k)$. According to Eq. (7), we recall that

$$q_n(k) = \frac{1}{k} \sum_{i=1}^{k} \mathcal{K}\left(\frac{d(\omega_n, \omega_n^i)}{h}\right). \tag{20}$$

Consider the difference,

$$
\begin{aligned}
q_n(k+1) - q_n(k) &= \frac{1}{k+1} \sum_{i=1}^{k+1} \mathcal{K}\left(\frac{d(\omega_n, \omega_n^i)}{h}\right) - \frac{1}{k} \sum_{i=1}^{k} \mathcal{K}\left(\frac{d(\omega_n, \omega_n^i)}{h}\right) \\
&= \frac{1}{k+1}\left[\mathcal{K}\left(\frac{d(\omega_n, \omega_n^{k+1})}{h}\right) - \frac{1}{k} \sum_{i=1}^{k} \mathcal{K}\left(\frac{d(\omega_n, \omega_n^i)}{h}\right)\right].
\end{aligned}
\tag{21}
$$

Since the neighbors are ordered in ascending distance, we have $d(\omega_n, \omega_n^{k+1}) \geq d(\omega_n, \omega_n^k)$. Because $\mathcal{K}(\cdot)$ is non-increasing kernel function, the $(k+1)$-th kernel value is upper bounded above by each of the first $k$ kernel values, and hence by their average. Therefore,

$$q_n(k+1) - q_n(k) \leq 0. \tag{22}$$

This establishes the monotonic non-increasing property of the client-level density $q_n(k)$.

Second, we analyze source-level monotonicity of density $q^s(k)$. According to Eq. (9), the source-level density is given by

$$q^s(k) = \frac{1}{N^s} \sum_{n \in \mathcal{N}_s} \frac{q_n(k)}{k}. \tag{23}$$

Consider the difference,

$$q^s(k+1) - q^s(k) = \frac{1}{N^s} \sum_{n \in \mathcal{N}_s} \frac{q_n(k+1)}{k+1} - \frac{1}{N^s} \sum_{n \in \mathcal{N}_s} \frac{q_n(k)}{k}. \tag{24}$$

Since $q_n(k)$ is monotonically non-increasing for each client $n$, each term in the summations satisfies $q_n(k+1) - q_n(k)$. Hence,

$$q^s(k+1) - q^s(k) \leq 0. \tag{25}$$

Therefore, the source-level density $q^s(k)$ is also monotonically non-increasing, which completes the proof.

## A.3. Proof of Theorem 4.1

According to the assumption A.1, when the decision variable $k$ lies in the region $k < \hat{k}$, the density of of cluster $s^*$ is no smaller than that of any other cluster. By Lemma A.2, $q^s(k)$ is monotonically non-increasing w.r.t. $k$. The following cases exhaust all possibilities. Therefore, we following three cases for $q^s(k)$ among different sources $s \in \mathcal{S}$ and different $k \in [1, k] \cap \mathbb{Z}$.

*Case 1.* For all $k$, $q^{s*}(k) > q^s(k)$.

*Case 2.* There exists a unique threshold $\hat{k}$ such that, for all $k$,

$$\begin{cases} q^{s*}(k) > q^s(k), & \text{if } k < \hat{k}, \\ q^{s*}(k) = q^s(k), & \text{if } k = \hat{k}, \\ q^{s*}(k) < q^s(k), & \text{if } k > \hat{k}. \end{cases} \tag{26}$$

*Case 3.* There exist multiple thresholds $\{\hat{k}_1, \cdots, \hat{k}_M\}$ such that

$$q^{s*}(\hat{k}_m) = q^s(\hat{k}_m), m \in [1, M] \cap \mathbb{Z}, \tag{27}$$

and the ordering alternates accordingly between successive thresholds.

Due to the monotonic non-increasing property established in Lemma A.2, in all cases the objective function admits a unique optimal value with respect to the decision variable $k$. This completes the proof.

## B. Proof of Theorem 5.2

To analyze the impact of noisy local updates on the margin, we focus on the worst-case scenario where noise is injected into the data source with the largest density, or equivalently, the most data-balanced source. This choice is aligns with our margin-based design objective, which assigns larger aggregation weights to more data-balanced sources. Consequently, noise affecting such a source has the most significant influence on the margin.

First, we derive the margin under clean and noisy settings, respectively. Let the margin under clean conditions be defined as

$$m = q^{s(1)} - q^{s(2)}, \tag{28}$$

where $q^{s(1)}$ and $q^{s(2)}$ denote the largest and second-largest density values among all data sources, respectively. After introducing noise, the margin becomes

$$m' = q^{s(1)'} - q^{s(2)}, \tag{29}$$

where $q^{s(1)'}$ represents the density of the largest source under noisy local updates. Accordingly, the margin loss induced by noise is

$$\Delta = m - m' = q^{s(1)} - q^{s(1)'}. \tag{30}$$

For notational simplicity, we denote

$$\Delta = q^s - q^{s'}, \tag{31}$$

where $q^s$ and $q^{s'}$ correspond to the clean and noisy density estimates of the same source, respectively.

Second, we decompose the margin into two parts: one part corresponds to the case where the noisy client serves as the center client, and the other part corresponds to the case where the noisy client appears in other clients' neighborhoods. Recall that the density estimate for data source $s$ is defined as

$$q^s = \frac{1}{N^s} \sum_{j \in \mathcal{N}_s} \frac{1}{k} \sum_{i=1}^{k} \mathcal{K}\left(\frac{d(\omega_j, \omega_j^i)}{h}\right), \tag{32}$$

where $\mathcal{K}(\cdot)$ is the kernel function with bandwidth $h$, and $\{\omega_j^i\}_{i=1}^k$ denote the neighborhood models of client $j$. Without loss of generality, assume that noise affects client $n \in \mathcal{N}_s$. Then $q^s$ can be decomposed as

$$q^s = \frac{1}{\mathcal{N}^s}\left[\frac{1}{k} \sum_{i=1}^{k} \mathcal{K}\left(\frac{d(\omega_n, \omega_n^i)}{h}\right) + \sum_{j \in \mathcal{N}_s/\{n\}} \frac{1}{k} \sum_{i=1}^{k} \mathcal{K}\left(\frac{d(\omega_j, \omega_j^i)}{h}\right)\right]. \tag{33}$$

By explicitly separating kernel terms involving client $n$ from those contributed by the remaining clients, we can isolate the influence of local noise on the density estimate. Let $\omega_n$ denote the clean local model of client $n$, and $\omega_n'$ denote its noisy counterpart. The difference between clean and noisy density estimates can thus be written as

$$\Delta = \frac{1}{N^s}\frac{1}{k}\{\sum_{i=1}^{k}[\mathcal{K}(\frac{d(\omega_n,\omega_n^i)}{h}) - \mathcal{K}(\frac{d(\omega_n',\omega_n^i)}{h})] + \sum_{j\in\mathcal{N}_s/\{n\}}[\mathcal{K}(\frac{d(\omega_j,\omega_n)}{h}) - \mathcal{K}(\frac{d(\omega_j,\omega_n')}{h})]\}. \tag{34}$$

Third, we provide a bound on the kernel perturbation. By the triangle inequality, the absolute deviation can be upper-bounded as:

$$|\Delta| \le \frac{1}{N_s k}\sum_{i=1}^{k}|\mathcal{K}(\frac{d(\omega_n,\omega_n^i)}{h}) - \mathcal{K}(\frac{d(\omega_n',\omega_n^i)}{h})| + \sum_{j\in\mathcal{N}_s/\{n\}}|\mathcal{K}(\frac{d(\omega_j,\omega_n)}{h}) - \mathcal{K}(\frac{d(\omega_j,\omega_n')}{h})|. \tag{35}$$

We assume that the local model perturbation induced by noise is bounded, i.e.,

$$\| \omega_n - \omega_n' \| \le \beta, \tag{36}$$

which implies

$$\frac{d(\omega_n,\omega_n')}{h} \in [0, \frac{\beta^2}{h}]. \tag{37}$$

Let the kernel function be Gaussian, i.e., $\mathcal{K}(x) = \frac{1}{(2\pi)^{r/2}}\exp(-x)$. Then

$$\mathcal{K}(\frac{d(\omega_n,\omega_n')}{h}) \in [\frac{1}{(2\pi)^{r/2}}\exp(-\frac{\beta^2}{h}), \frac{1}{(2\pi)^{r/2}}]. \tag{38}$$

As a result, the absolute difference between two kernel evaluations satisfies

$$|\mathcal{K}(\frac{d(\omega_j,\omega_n)}{h}) - \mathcal{K}(\frac{d(\omega_j,\omega_n')}{h})| \le \frac{1 - \exp(-\beta^2/h)}{(2\pi)^{r/2}}. \tag{39}$$

Applying this bound to both summation terms yields

$$\sum_{i=1}^{k}|\mathcal{K}(\frac{d(\omega_n,\omega_n^i)}{h}) - \mathcal{K}(\frac{d(\omega_n',\omega_n^i)}{h})| \le \frac{K(1 - \exp(-\beta^2/h))}{(2\pi)^{r/2}}, \tag{40}$$

and

$$\sum_{j\in\mathcal{N}_s/\{n\}}|\mathcal{K}(\frac{d(\omega_j,\omega_n)}{h}) - \mathcal{K}(\frac{d(\omega_j,\omega_n')}{h})| \le \frac{(N_s-1)(1 - \exp(-\beta^2/h))}{(2\pi)^{r/2}}. \tag{41}$$

Combining both terms, we obtain

$$|\Delta| \le \frac{(K + N_s - 1)[1 - \exp(-\frac{\beta^2}{h})]}{N^s(2\pi)^{r/2}} = \xi_n \tag{42}$$

Since the margin estimator is a sum of independent bounded kernel evaluations, we can apply McDiarmid's inequality (Long et al., 2024; McDiarmid, 1989) to characterize its deviation from the clean expectation. Specifically,

$$\mathbb{P}(|m - \mathbb{E}[m]| \ge \gamma) \le \exp(-\frac{2\gamma^2}{\sum_{n=1}^{N}\xi_n}), \tag{43}$$

which completes the proof.

## C. Proof of Lemma 5.3

According to Eq. (14), we recall the update rule at round $k$ as

$$\hat{m}_k = \hat{m}_k^- + \kappa_k(\tilde{m}_k - \hat{m}_k^-), \tag{44}$$

where $\hat{m}_k^-$ denotes the prior estimate and $\tilde{m}_k$ denotes the observed margin at round $k$. The corresponding estimation error can be written as

$$\begin{aligned} e_k &= \hat{m}_k - m_k \\ &= (1 - \kappa_k)(\hat{m}_k - m_k) + \kappa_k(\tilde{m}_k - m_k) \\ &= (1 - \kappa_k)e_k^- + \kappa_k v_k, \end{aligned} \tag{45}$$

where $e_k^- = \hat{m}_k^- - m_k$ denotes the prior estimation error and $v_k = \tilde{m}_k - m_k$ denotes the observation noise. We assume that $e_k^-$ and $v_k$ are independent zero-mean random variables with variances

$$\begin{cases} \mathrm{Var}(e_k^-) = p_k^-, \\ \mathrm{Var}(v_k) = \beta. \end{cases} \tag{46}$$

This independence implies that the posterior estimation error variance is given by

$$\begin{aligned} p_k &= \mathrm{Var}(e_k) \\ &= (1 - \kappa_k)^2 \mathrm{Var}(e_k^-) + \kappa_k^2 \mathrm{Var}(v_k) \\ &= (1 - \kappa_k)^2 p_k^- + \kappa_k^2 \beta. \end{aligned} \tag{47}$$

To obtain the optimal Kalman gain $\kappa_k$, we minimize $p_k$ with respect to $\kappa_k$. Taking the derivative yields

$$\frac{\partial p_k}{\partial \kappa_k} = -2(1 - \kappa_k)p_k + 2\kappa_k\beta. \tag{48}$$

Moreover, the second-order derivative is

$$\frac{\partial^2 p_k}{\partial \kappa_k^2} = 2(p_k + \beta) > 0. \tag{49}$$

which indicates that $p_k$ is strictly convex function of $\kappa_k$. Therefore, the optimal Kalman gain is obtained by setting the first-order derivative to zero, yielding

$$\kappa_k^* = \frac{p_k^-}{p_k^- + \beta}. \tag{50}$$

The proof is complete.

## D. Proof of Theorem 5.4

Let $m_k$ denote the true value at round $k$, and let $\tilde{m}_k$ be a noisy observation. Define the measurement noise as $v_k = m_k - \tilde{m}_k$. According to Theorem 5.1, the noise is bounded such that $|v_k| \le \xi$. According to Eq. (14) , we reformulate the Kalman update rule as

$$\hat{m}_k = (1 - \kappa_k)\hat{m}_k^- + \kappa_k\tilde{m}_k. \tag{51}$$

Define the prior error and estimate error as

$$\begin{cases} e_k^- = m_k - \hat{m}_k^-, \\ e_k = m_k - \hat{m}_k. \end{cases} \tag{52}$$

Substituting the update rule into the error definition yields

$$e_k = (1 - \kappa_k)e_k^- + \kappa_k v_k. \tag{53}$$

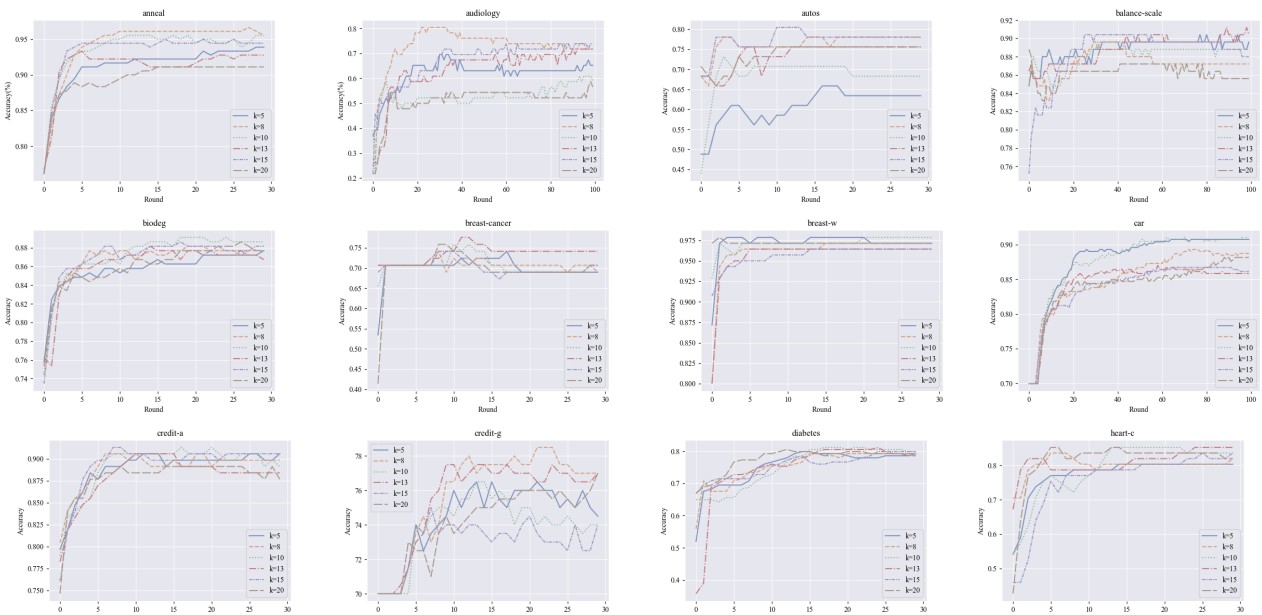

*Figure 6.* Accuracy comparison across different neighborhood sizes on industry-oriented datasets from CEKA toolkit.

Taking absolute values and using the triangle inequality, we obtain

$$|e_k| \leq (1 - \kappa_k)|e_k^-| + \kappa_k|v_k|$$
$$\leq (1 - \kappa_k)|e_k^-| + \kappa_k \xi. \tag{54}$$

Since $(1 - \kappa_k)$ and $\kappa_k$ are non-negative and sum to one, the right-hand side is a convex combination of $|e_k^-|$ and $\xi$. Therefore,

$$|e_k| \leq \max\{|e_k^-|, \xi\}. \tag{55}$$

This bound guarantees that the Kalman update does not amplify the worst-case error beyond the larger of the prediction error and the measurement error bound. We have $\text{Var}(e_k^-) = p_k^-$, $\text{Var}(e_k) = p_k$, and $\text{Var}(v_k) = \beta,$. Substituting Eq. (15) into Eq. (16), we have

$$p_k = \frac{\beta p_k^-}{p_k^- + \beta}. \tag{56}$$

As a result, we have $p_k < p_k^-$ and $p_k < \beta$, which shows that the Kalman update strictly reduces the estimation uncertainty in the mean-square sense. Although the Kalman filter minimizes mean-square error rather than absolute error, the latter can be upper bounded using the Cauchy-Schwarz inequality:

$$\mathbb{E}|e_k| \leq \sqrt{\mathbb{E}[e_k^2]}. \tag{57}$$

According to $\mathbb{E}[e_k^2] = \text{Var}(e_k) = p_k$, and hence

$$\mathbb{E}|e_k| \leq \sqrt{p_k} = \sqrt{\frac{\beta p_k^-}{p_k^- + \beta}} < \sqrt{\beta}. \tag{58}$$

Finally, since the measurement noise is bounded by $\xi$, we have $\beta = \mathbb{E}[v_k^2] \leq \xi^2$. Consequently,

$$\mathbb{E}|e_k| < \xi. \tag{59}$$

Therefore, we let $\zeta = \sqrt{\frac{\beta p_k^-}{p_k^- + \beta}}$, and $\zeta < \xi$. According to McDiarmid's inequality, we have

$$\mathbb{P}(|m - \mathbb{E}[m]| \geq \gamma) \leq \exp(-\frac{2\gamma^2}{\sum_{n=1}^N \zeta}). \tag{60}$$

This completes the proof.

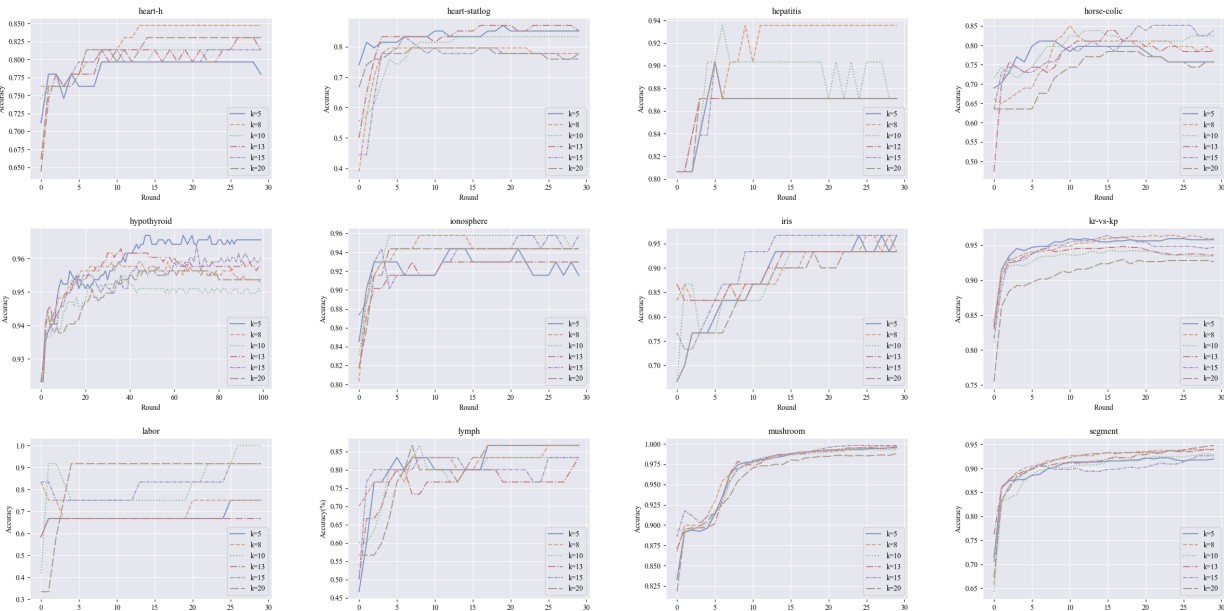

*Figure 7.* Accuracy comparison across different neighborhood sizes on industry-oriented datasets from CEKA toolkit.

# E. Supplement of Experiements

## E.1. Datasets and Experimental Setup

For MNIST (LeCun et al., 2021), Fashion-MNIST (Xiao et al., 2017), and FEMNIST (Caldas et al., 2018), we employ a 4-layer convolutional neural network (CNN) models. The architecture consists of two $5 \times 5$ convolutional layers with $10$ and $20$ channels, respectively, each followed by $2 \times 2$ max-pooling and ReLU activation, and two fully connected layers with output dimensions of $50$ and the number of classes. For CIFAR-10 (Krizhevsky, 2009), we adopt a $4$ layer CNN backbone containing three $3 \times 3$ convolutional layers with $32$, $64$, and $128$ channels, each followed by max-pooling, and a two-layer fully connected classifier with $256$ hidden units and ReLU activation. MNIST, Fashion-MNIST, FEMNIST, and CIFAR-10 are used to train local models for $1$ epoch with a batch size of $64$ across $60$ clients. The learning rate is set to $0.01$ for MNIST, Fashion-MNIST, and CIFAR-10, and $0.005$ for FEMNIST.

For 34 industry-oriented datasets from CEKA toolkit, we clean ARFF format and apply standard tabular data processing to ensure compatibility with federated Multi-Layer Perceptron (MLP) training and unified evaluation, before training. We adopt a 3-layer fully connected MLP classifier with hidden dimensions $128$ and $64$. Each hidden layer uses ReLU activation and applies Dropout with a rate of $0.1$ for regularization. Local training is performed for $2$ epochs with a batch size of $64$, and the learning rate is set to $0.001$.

To induce data skew, we partition all clients into two groups: 10 non-skew clients and 50 skew clients. For label skew, we impose imbalanced class proportions on skew clients. For feature skew, skew clients apply distinct data transforms on their training data.

## E.2. Baseline Description

We compare FedVeer with four baseline methods:

- LoMar (Li et al., 2021): A local defense against poisoning attacks in FL that employs kernel density estimation over a fixed-size neighborhood, under the assumption that only a minority of clients launch poisoning attack.

- FedLC (Zhang et al., 2022): An FL algorithm that mitigates label distribution skew by logits calibration using client local label statistics, without concern feature skew.

- FedRDN (Yan et al., 2025): A data augmentation based FL method that mitigates feature distribution skew by randomly

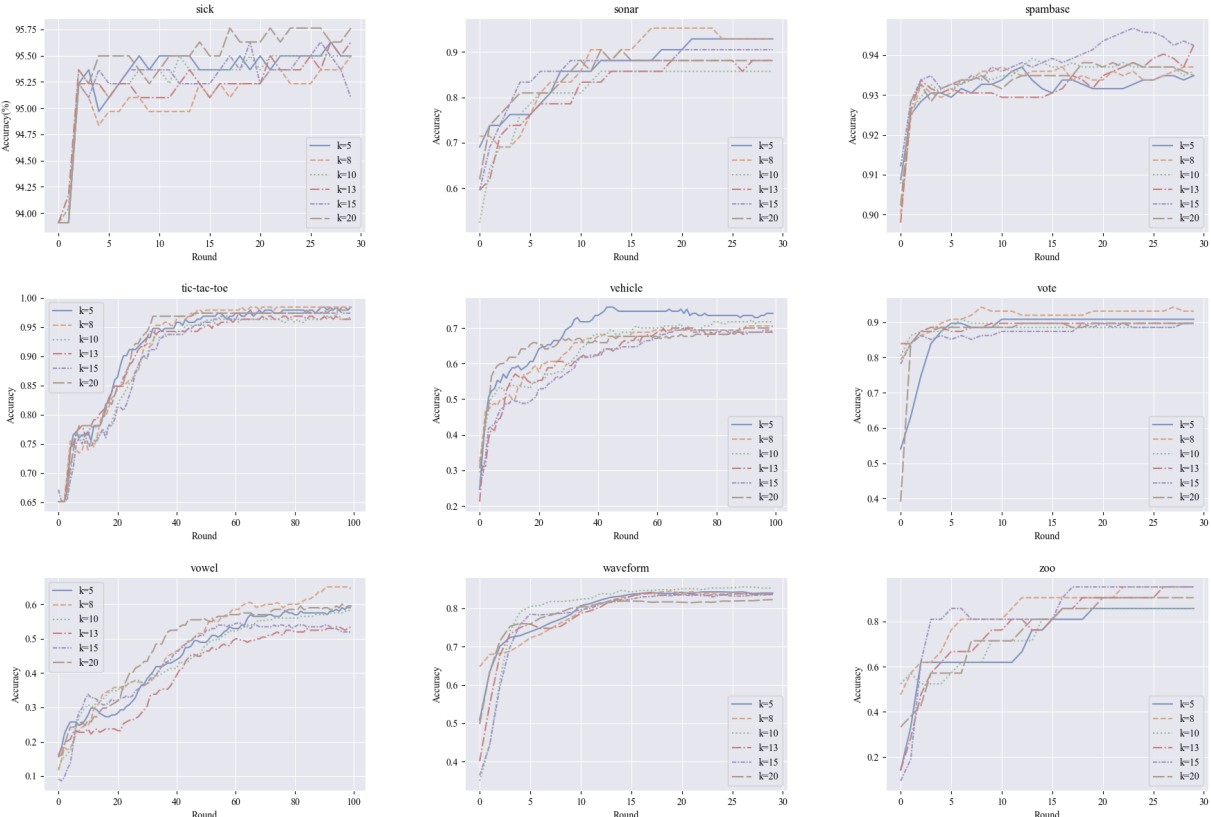

*Figure 8.* Accuracy comparison across different neighborhood sizes on industry-oriented datasets from CEKA toolkit.

injects local distribution statistics from the entire federated into the client's data, without addressing label distribution skew.

- TPFL (Gu et al., 2026): This method relies on a dominance-based density assumption in gradient space and employs a two-dimensional detection mechanism that combines kernel density estimation with directional consistency analysis to identify malicious gradients under poisoning attacks.

- FedLSA (Fu et al., 2025): This method addresses domain skew through representation learning with learnable semantic anchors and hyperspherical contrastive embeddings to improve feature separability across heterogeneous clients.

### E.3. Summlement Experiment Results

In Figures 6–8, we observe that different neighborhood sizes lead to varying convergence speeds and final accuracies on industry-oriented datasets from the CEKA toolkit. These results indicate that adaptively adjusting the neighborhood size is crucial for achieving faster convergence and higher accuracy. An appropriate neighborhood size enables more balanced clients to attain higher density estimates and, consequently, larger aggregation weights. In contrast, an inappropriate neighborhood size may increase the weights of skewed clients, thereby amplifying the skewness of the global model, which becomes particularly pronounced when majority clients exhibit highly skewed data distributions.

