# OpenReview forum: "FedVeer: Self-Adaptive Skew Estimation for Robust Federated Learning"
_ICML.cc/2026/Conference — ICML 2026 regular_

### Official Review · Reviewer_RhZ6 · 2026-03-09

**Soundness:** 3
**Presentation:** 3
**Significance:** 3
**Originality:** 3
**Overall Recommendation:** 3
**Confidence:** 3

**Summary:**

This paper introduces FedVeer, a robust federated learning framework designed to address the challenges of non-IID data distributions and stochastic noise in model updates. The framework moves away from traditional kernel density estimation methods that rely on fixed neighborhood structures. Instead, it employs a self-adaptive kernel density estimation module that utilizes max-margin learning to dynamically determine the optimal neighborhood size $k$. To mitigate the impact of noise-induced perturbations, FedVeer incorporates a Kalman filter-based stabilization mechanism to refine the margin estimation. The authors provide theoretical guarantees on margin stability and demonstrate the effectiveness of their approach through extensive experiments on both standard benchmarks and a large set of industry-oriented datasets.

**Compliance With Llm Reviewing Policy:**

Affirmed.

**Key Questions For Authors:**

Please refer to the section above on weaknesses.

**Limitations:**

yes

**Strengths And Weaknesses:**

Strengths:

1) The paper correctly identifies that in skewed federated environments, standard kernel-based estimates can be dominated by skewed majority clients, leading to biased model aggregation.

2) The integration of classical control theory with modern federated learning optimization is well-motivated and supported by rigorous theoretical analysis, including proofs for the uniqueness of the max-margin solution and tightened deviation bounds.

3)  The framework is validated across a diverse range of 34 industry-oriented datasets from the CEKA toolkit, in addition to standard image classification benchmarks, demonstrating strong generalization capability across different domains such as finance and healthcare.

Weaknesses:

1) The core mechanism of FedVeer assumes the existence of $S$ distinct data sources and relies on the calculation of source-level density $q^s$ to weight clients. In many real-world decentralized FL scenarios, the boundaries between data sources are often unknown or blurred, and the server may not have access to such group-level labels, which could limit the framework's practical deployment. Could the heavy use of data priors negatively impact privacy? A rigorous privacy proof is required to justify the security claims.

2)  While the method adaptively selects the neighborhood size $k$, it introduces several other sensitive hyperparameters, such as the Kalman filtering forgetting factor $\lambda$, process noise variance $\alpha$, and observation noise variance $\beta$. The paper lacks a detailed discussion on how to adaptively optimize these parameters in a fully unlabeled or highly dynamic environment.

3) The proposed max-margin learning algorithm requires the server to traverse the neighborhood size $k$ from 1 to $N$ to find the optimal value. In scenarios with a very large number of participating clients ($N$), this iterative search process could significantly increase the computational burden per aggregation round.

4) The effectiveness of kernel density estimation is highly dependent on the choice of the bandwidth parameter $h$. The authors do not provide a mechanism for the self-adaptive adjustment of $h$ when dealing with different model architectures or varying parameter dimensions, which might affect the robustness of the skew estimation.

---

> ### Author Rebuttal · Authors · 2026-03-30
>
> >**W1: Data source assumption and privacy concerns limit practical applicability.**
>
> **R1**: Thank you for raising concerns on applicability and privacy.
>
> (1) **No data source assumption**. FedVeer is fully **source-agnostic** and does not require known or explicit data source labels; “source” is only a conceptual abstraction of heterogeneity. In practice, all computations are performed in parameter space: client relationships are inferred via kernel density estimation over model updates, and aggregation weights are derived from continuous density margins. This avoids any reliance on predefined groups and naturally supports real-world cross-silo and cross-device FL, where source boundaries are unknown or blurred.
>
> (2) **Privacy**. FedVeer introduces no additional privacy risk beyond standard FL, as it operates solely on shared model updates without accessing raw data or metadata. The density-based weighting is functional rather than identificational and does not recover or expose group information. While a formal privacy proof is beyond the current scope, the framework is fully compatible with secure aggregation and differential privacy mechanisms.
>
> ---
>
> >**W2: Sensitivity of additional hyper-parameters and lack of adaptive tuning strategy.**
>
> **R2**: Thank you for raising this concern. The Kalman hyper-parameters ($\lambda$, $\alpha$, $\beta$) mainly serve a stabilizing role, while the core of FedVeer lies in density-based adaptive neighborhood selection.
>
> All parameters are estimated in a data-driven, label-free manner: $\beta$ from within-round update variance, $\alpha$ from inter-round variations, and $\lambda$ from recent update fluctuations to balance adaptivity and stability.
>
> To validate robustness, we vary $\beta$ across four datasets and measure the improvement ratio
> $\Delta = \frac{|m-\tilde{m}| - |m - \hat{m}|}{|m-\tilde{m}|}$, where $m$, $\tilde{m}$, and $\hat{m}$ denote clean, noisy, and filtered margins.
>
> |$\beta$|mnist|fashion-mnist|femnist|cifar-10|
> |:-----:|:------:|:-------------:|:-------:|:--------:|
> | 0.01  | -0.039 | 0.294         | 0.280   | 0.065    |
> | 0.02  | 0.019  | 0.370         | 0.239   | 0.204    |
> | 0.05  | 0.127  | 0.457         | 0.251   | 0.361    |
> | 0.10  | 0.227  | 0.489         | 0.267   | 0.512    |
>
> Larger $\Delta$ indicates greater error reduction. The results show consistent improvements across noise levels, with controlled degradation as $\beta$ increases. We will clarify these adaptive strategies and provide practical guidelines in the revision.
>
> ---
>
> >**W3: High overhead from searching over $k\in[1,N]$ when $N$ is large.**
>
> **R3**: Thank you for raising this concern. FedVeer does not require exhaustive search over $k\in[1,N]$.
>
> In Algorithm 1 (line 7), the search supports early stopping based on the density margin. For small $k$, the highest-density source remains stable; as $k$ increases, once this source changes, the margin no longer improves and typically decreases (Theorem 4.1), indicating that further expansion is unlikely to be beneficial. This avoids evaluating large $k$.
>
> As a result, the effective search is limited to $k\le \hat{k}$ with $\hat{k}\ll N$, yielding per-round complexity $O(\hat{k})$ instead of $O(N)$. Empirically, $\hat{k}$ remains small across datasets and the search converges within a few steps, incurring negligible overhead. We will further clarify this mechanism in the revision.
>
> ---
>
> >**W4: Sensitivity to bandwidth $h$ and lack of adaptive selection mechanism.**
>
> **R4**: Thank you for highlighting the importance of the bandwidth parameter $h$. While $h$ controls density smoothness and the bias–variance trade-off, its impact is mitigated by density margins and adaptive neighborhood selection.
>
> Our method relies on relative density comparisons, for which the ordering is largely preserved within a reasonable range of $h$. Moreover, the neighborhood size $k$ plays a more dominant role in determining locality, while $h$ mainly affects smooth weighting.
>
> Empirically, we evaluate $h\in${$x, x/2, x/4$}, where $x$ is the median pairwise distance between model updates. The results show that for each dataset, the performance exhibits a consistent trend with respect to $k$, regardless of the choice of $h$.
> | dataset | anneal (k=5) | anneal (k=8) | anneal (k=13) | audiology (k=5) | audiology (k=8) | audiology (k=13) |
> |---------|-------------|-------------|--------------|----------------|----------------|-----------------|
> | h = x   | 0.9389      | 0.9611      | 0.9389       | 0.7391         | 0.8043         | 0.6739          |
> | h = x/2 | 0.9389      | 0.9556      | 0.9389       | 0.6957         | 0.7826         | 0.6304          |
> | h = x/4 | 0.9222      | 0.9500      | 0.9389       | 0.6087         | 0.7174         | 0.6087          |
>
> Regarding adaptivity, $h$ is set using the median pairwise distance as a robust, scale-aware default. We will clarify this in the revision.

---

### Official Review · Reviewer_B3b9 · 2026-03-11

**Soundness:** 4
**Presentation:** 3
**Significance:** 4
**Originality:** 4
**Overall Recommendation:** 5
**Confidence:** 5

**Summary:**

This paper proposes a skew-aware FL framework that leverages self-adaptive kernel density estimation to mitigate data skew arising from fixed neighborhood structures and noise-induced perturbations in the kernel space. The neighborhood size is dynamically optimized via a max-margin learning to reduce majority-client skew, while Kalman filtering is incorporated to stabilize margin estimation under noisy updates. Theoretical contributions include a proof of the uniqueness of the optimal neighborhood size and a high-probability upper bound on margin deviation, which is shown to be tightened after filtering. Extensive experiments on multiple benchmark and real-world datasets demonstrate consistent performance gains over existing baselines under both data skew and noise conditions.

**Compliance With Llm Reviewing Policy:**

Affirmed.

**Final Justification:**

After reviewing the rebuttal and other comments, my evaluation remains unchanged. The authors have addressed prior concerns, demonstrating that the source-level margin learning method effectively mitigates source misalignment and adapts to client-side data skew. Its online, parameter-free design supports robustness and practicality. Minor issues do not affect the core contributions. I maintain Accept.

**Key Questions For Authors:**

1.Why is adaptive neighborhood size learning based only on source-level density, while both client- and source-level densities are used for aggregation weights?
2.How does the framework perform under dynamic client availability? Is it robust to significant fluctuations in client participation across rounds?
3.Under what real-world skew conditions does Assumption A.1 (most balanced source yields highest density) hold? Have you observed cases where this ordering becomes unstable?
4.What is the intuition or empirical evidence linking density margin maximization to improved global model performance?

**Limitations:**

FedVeer guarantees a unique optimal neighborhood size under Assumption A.1, which assumes a clear density separation among data sources. While this assumption holds in many heterogeneous settings, it may be violated when most clients exhibit highly aligned skew in the same direction, potentially undermining the theoretical guarantee. Furthermore, the Kalman filtering mechanism models margin estimation error as unbiased noise, an assumption that may not capture more complex or structured dynamics in real federated systems. Extending the robustness analysis to accommodate broader classes of noise models would enhance the framework's practical applicability.

**Strengths And Weaknesses:**

Strengths:
•Theoretical guarantees and experimental validation: The paper provides a rigorous theoretical analysis, including a proof of the uniqueness of the optimal neighborhood size in the proposed max-margin learning problem. It further establishes a high-probability upper bound on the learning deviation and shows that this bound can be tightened through Kalman filtering. These theoretical claims are well supported by experimental results, which systematically evaluate the impact of neighborhood size and noise on margin learning performance.
•Clarity and structural coherence: The paper clearly articulates the concept of data skew, the limitations of existing approaches in handling it, and the technical challenges addressed by the proposed framework. The presentation follows a logical progression: it first formulates the problem of kernel density estimation under majority-client skew in noisy setting, then proposes a noise-free version of FedVeer that adaptively determines the neighborhood size for kernel density estimate via max-margin learning, and finally extends it to a noise-robust version of FedVeerthat incorporates a customized Kalman filtering mechanism to mitigate noise-induced perturbations.
•Practical impact and methodological contribution: The proposed framework enables skew estimation via kernel-based methods without requiring access to local features, labels, or their statistics, which is a significant advantage in privacy-preserving federated settings. By dynamically adjusting the neighborhood size through max-margin learning, it mitigates the amplification effect caused by skewed majority clients. Moreover, the integration of Kalman filtering into the max-margin learning process enhances robustness against noisy updates, addressing a key practical challenge in real-world deployments.
•Novel kernel-based adaptive mechanism: A core innovation lies in the adaptive selection of the neighborhood size in kernel density estimation, which is optimized via a max-margin formulation without relying on fixed priors. This mechanism reduces bias toward skewed majority clients and improves the accuracy of skew estimation. The incorporation of Kalman filtering further stabilizes the learning process under noise, making the approach more resilient and applicable to realistic federated learning scenarios.

Weakness:
•While the framework incorporates both client-level and source-level kernel density estimations for aggregation weight assignment, the max-margin learning process is applied solely to the source-level density. The rationale behind this asymmetric design and its potential implications for overall skew mitigation are not fully discussed.
•The effectiveness of the max-margin learning mechanism is contingent upon the availability of clients for kernel density estimation. However, the paper does not adequately address scenarios where client availability fluctuates dynamically across rounds. The robustness of the proposed approach under such practical conditions remains unclear.
•The theoretical guarantee regarding the uniqueness of the optimal neighborhood size relies on the assumption that the most balanced source consistently attains the highest kernel density. While this assumption is intuitively plausible, its validity may not hold in settings with complex or overlapping skew patterns. A more detailed discussion of the conditions under which this assumption is justified would help clarify the practical scope of the theoretical result.
•The neighborhood size is selected by maximizing the density margin, yet the connection between this objective and expected loss minimization is not explicitly established. Clarifying how margin maximization relates to final model performance would strengthen the conceptual foundation of the approach.

---

> ### Author Rebuttal · Authors · 2026-03-30
>
> >**Q1&W1: Asymmetry use of density levels in adaptive learning vs. aggregation.**
>
> **R1**: We appreciate this thoughtful question. Data heterogeneity in FL arises at two levels: (i) source-level skew, where entire groups are misaligned, and (ii) client-level skew, reflecting finer local deviations.
>
> Accordingly, we adopt a hierarchical and asymmetric design. At the source level, max-margin learning is used to identify and filter globally misaligned sources, where separability is more plausible. At the client level, density-based reweighting provides a continuous and stable adjustment to suppress locally skewed clients.
>
> Max-margin learning is not applied at the client level, as local variations are typically non-separable and would introduce instability.
>
> This asymmetry reflects the distinct statistical structures at each level, enabling reliable global selection and fine-grained local refinement.
>
> ---
>
> >**Q2&W2: Insufficient Handling of Dynamic Client Availability.**
>
> **R2**: Thank you for pointing out this practical issue. Our method naturally accommodates dynamic client availability, as all computations are performed on the set of active clients in each round. Both kernel density estimation and margin learning are recomputed online, without assuming a fixed client pool.
>
> Crucially, the method relies on relative density comparisons rather than absolute values, making it inherently robust to fluctuations in participation. As the active set changes, the density landscape is updated accordingly, and the neighborhood size $k$ is adaptively re-estimated from the current distribution.
>
> This design ensures stable identification of aligned sources and suppression of skewed clients under dynamic participation, supporting robust performance in practical FL scenarios.
>
> ---
>
> >**Q3&W3: Limited validity of Assumption A.1.**
>
> **R3**: Thank you for this valuable comment. We discuss this assumption in detail in the appendix (Remark A.1). Intuitively, clients with better distribution alignment tend to produce more consistent update directions, leading to higher kernel density, while misaligned clients generate more dispersed updates with lower density. This principle is also widely used in anomaly detection and similarity-based methods [1–3].
>
> The assumption is most valid when data within each source is relatively consistent and different sources exhibit clear distributional distinctions, as in our setting. In such cases, kernel density effectively reflects alignment in the update space.
>
> We acknowledge that under highly overlapping or complex skew, the assumption may weaken. However, these scenarios inherently lack separability, making reliable source discrimination difficult for any method. Our framework mitigates this by combining source-level density estimation with client-level reweighting, improving robustness in practice.
>
> [1] Luo, J. and Wu, S. Fedsld: Federated Learning with Shared Label Distribution for Medical Image Classification. In Proceedings of the 19th IEEE International Symposium on Biomedical Imaging (ISBI 2022), Kolkata, India, 2022. IEEE.
>
> [2] Li, X., Qu, Z., Zhao, S., Tang, B., Lu, Z., and Liu, Y. LoMar: A Local Defense Against Poisoning Attack on Federated Learning. IEEE Transactions on Dependable and Secure Computing, 20(1):437–450, 2021.
>
> [3] Cao, X., Zhang, Z., Jia, J., and Gong, N. Z. FLCert: Prov- ably Secure Federated Learning Against Poisoning At- tacks. IEEE Transactions on Information Forensics and Security, 17:3691–3705, 2022.
>
> ---
>
> >**Q4&W4: Unclear link between density margin maximization and model performance.**
>
> **R4**: We thank you for this thoughtful comment.
>
> The density margin objective aims to separate well-aligned and misaligned updates in the model update space. Higher density corresponds to more consistent update directions, while lower density indicates dispersed and potentially skewed updates. Maximizing this margin therefore improves the distinguishability between reliable and unreliable clients.
>
> This enhanced separability leads to more accurate neighborhood selection, allowing aggregation to focus on well-aligned updates and reducing the influence of skewed ones. As a result, the aggregated direction better approximates the true optimization trajectory.
>
> Consequently, maximizing the density margin yields more stable aggregation, improving convergence and reducing training loss.

---

> > ### Author Rebuttal · Reviewer_B3b9 · 2026-04-01
> >
> > The authors have effectively resolved my earlier concerns. The proposed source-level margin learning approach successfully filters globally misaligned sources, especially in the presence of significant client-side data skew. By maximizing the margin, the model achieves greater stability and improved performance. In addition, the framework operates without static parameters and inherently adapts to dynamic conditions through its online learning mechanism.

---

> > > ### Author Response · Authors · 2026-04-02
> > >
> > > We sincerely thank you for your positive assessment and for recognizing the effectiveness of our source-level margin learning approach. We greatly appreciate your thoughtful feedback throughout the review process. We are glad that our clarifications adequately addressed your concerns, and we will ensure that all related insights from the rebuttal are clearly reflected in the final version of the manuscript.

---

### Official Review · Reviewer_C8QP · 2026-03-11

**Soundness:** 3
**Presentation:** 3
**Significance:** 3
**Originality:** 2
**Overall Recommendation:** 4
**Confidence:** 3

**Summary:**

This paper proposes FedVeer, a self-adaptive skew estimation method for addressing the non-IID challenge in federated learning (FL). The authors identify two limitations of existing approaches: bias toward skewed majority clients during training and performance degradation caused by noise-induced perturbations. To address the first issue, the paper introduces a max-margin learning scheme to determine the optimal neighborhood size adaptively. To address the second, it incorporates Kalman filters to reduce the impact of noisy perturbations. The proposed method is evaluated on multiple datasets and compared against several baseline approaches. The results validate the effectiveness of the proposed mechanism.

**Compliance With Llm Reviewing Policy:**

Affirmed.

**Final Justification:**

The rebuttal has addressed my concerns.

**Key Questions For Authors:**

This is generally a good paper, and I do see its technical contributions. However, I am still not fully convinced by some of the design choices. In particular, the paper should provide more explanation for why the optimization objective is to maximize the margin between the largest and second-largest kernel densities, and why this objective is expected to improve performance.

**Limitations:**

Yes

**Strengths And Weaknesses:**

Strengths:
1. Interesting problem: The paper is well motivated. Addressing data skew is a fundamental and important challenge in federated learning.
2. Novel design: The adaptive kernel-based weighting scheme and Kalman filter are both reasonable design choices for addressing the identified challenges. The paper also provides a fairly rigorous analysis of the proposed mechanism.
3. Extensive experiments: The paper conducts extensive experiments to evaluate the proposed mechanism. The results demonstrate its effectiveness, support the design choices, and show that it outperforms existing baseline methods.

Weaknesses:
1. Needs more design intuitions: The paper needs to provide more explanation of the design rationale. In particular, I am not fully convinced why maximizing the margin between the largest and second-largest kernel densities would clearly lead to better performance. A separate discussion section would be helpful to connect this design choice to the design objective and the empirical results.
2. Minor technical flaws: For example, in Equation (10), the paper claims that summing over all n leads to 1, but this does not seem correct. Based on the formulation, the normalization appears to require summation over both n and s to equal 1.

---

> ### Author Rebuttal · Authors · 2026-03-30
>
> >**W1&Q1: Lack of clear design intuition and justification for the max-margin objective.**
>
> **R1:** We thank you for this insightful comment. You raise an important question about why maximizing the margin between the top two kernel densities leads to improved performance and how this design aligns with our objective. To address this, we clarify the intuition from both a modeling and optimization perspective, and explain how it connects to empirical gains:
>
> (1) **Margin maximization for robustness**: Our design follows the principle of margin maximization to improve robustness and decision reliability. Since raw data are unavailable in FL, we realize this idea via kernel density estimation over model updates.
>
> (2) **Density as update consistency**: In our framework, kernel density reflects how consistent client updates are within a source—well-aligned clients form concentrated, high-density regions. However, absolute density values are sensitive to local geometry, sampling, and neighborhood size, making them unreliable for cross-source comparison.
>
> (3) **Margin as a confidence measure**: We use the margin between the largest and second-largest densities as a measure of confidence. A large margin indicates clear alignment with one source, while a small margin signals ambiguity. This relative criterion is more discriminative and robust than absolute densities.
>
> (4) **Stability in optimization**: Maximizing this margin stabilizes client weighting. When the margin is small, minor perturbations (e.g., stochastic updates or noise) can flip the selected source, leading to unstable aggregation. A larger margin improves resilience, resulting in smoother updates and reduced variance across rounds.
>
> (5) **Empirical support**: This improved stability translates into better convergence behavior, which is empirically supported by our results (e.g., Table 1, Figure 4). We will further clarify this design intuition and its connection to empirical observations in the revision.
>
> ---
>
> >**W2: Normalization issue in Eq. (10): summation over $n$ alone does not yield 1.**
>
> **R2:** Thank you for carefully pointing this out.
>
> We agree that the original formulation may be confusing and could suggest that normalization requires summation over both $𝑛$ and $𝑠$. We clarify that the weights are defined at the client level and normalized globally, so the condition **$\sum_{n=1}^N \theta_n = 1$ holds**.
>
> The apparent dependence on $𝑠$ comes from aggregating source-level densities into a per-client quantity; it does not introduce an additional normalization dimension. To make this explicit and avoid ambiguity, we will revise Eq. (10) to the following normalized form:
>
> $$\theta_n = \frac{q_n\sum_{s=1}^S q^s\mathbf{1}[n\in s]}{\sum_{n=1}^N q_n\sum_{s=1}^S q^s\mathbf{1}[n\in s]},$$
>
> where $\mathbf{1}[n\in s]$ indicates whether client $𝑛$ is associated with source $𝑠$. Under this form, it is straightforward to verify that $\sum_{n=1}^N \theta_n = 1$.
>
> We will revise Eq. (10) and its explanation accordingly to make the normalization clear.

---

> > ### Author Rebuttal · Reviewer_C8QP · 2026-04-01
> >
> > Thanks for your response. I do not have any further concerns. Please incorporate the insights from your rebuttal into the manuscript.

---

> > > ### Author Response · Authors · 2026-04-02
> > >
> > > We sincerely thank you for your positive feedback and for confirming that your concerns have been fully resolved. We appreciate your constructive comments throughout the review process. We will carefully incorporate the insights and clarifications from our rebuttal into the revised manuscript to further improve its clarity and completeness.

---

### Official Review · Reviewer_rhsK · 2026-03-12

**Soundness:** 2
**Presentation:** 3
**Significance:** 2
**Originality:** 2
**Overall Recommendation:** 4
**Confidence:** 2

**Summary:**

This paper addresses the performance degradation of Federated Learning (FL) under non-IID data distributions, specifically focusing on the challenges posed by data skew. The authors investigate the fundamental limitations of existing kernel-based skew estimation methods, such as their bias toward majority-skewed clients and vulnerability to noise-induced perturbations. To mitigate these issues, the paper proposes FedVeer, a skew-aware FL framework that utilizes self-adaptive kernel density estimation with k-free neighborhoods. FedVeer dynamically determines optimal neighborhood sizes via max-margin learning to reduce bias and incorporates Kalman filtering to stabilize estimation under noisy updates. Extensive experiments demonstrate the effectiveness of the proposed method.

**Compliance With Llm Reviewing Policy:**

Affirmed.

**Final Justification:**

The article is enhanced by the detailed rebuttal and a recent baseline comparison. The innovation concern is partially resolved. So, I will increase my score to 4.

**Key Questions For Authors:**

Please refer to Weaknesses.

**Limitations:**

yes

**Strengths And Weaknesses:**

Strengths

- The manuscript is well-written and clearly structured.
- The study addresses a critical challenge of maintaining federated learning performance when the majority of clients exhibit skewed data.
- Extensive experiments conducted on multiple datasets demonstrate the effectiveness of the proposed method.


Weaknesses

- Although this paper integrates Kernel Density Estimation (KDE) with Kalman filtering to address the issue of data skew in FL, the novelty of its approach remains difficult to define. The algorithm adapts the self-adaptive neighborhood concept from KFNN to the FL setting and layers it onto the model parameter analysis framework already established by LoMar.
- The proposed scheme relies heavily on the assumption that the data source of each client is known a priori. In privacy-preserving FL scenarios, identifying or grouping clients by their specific data sources is often impractical.
- The scope of comparative analysis is too narrow. The selected baselines (FedAvg, LoMar, FedLC) focus primarily on early research results predating 2022.
- FedVeer comprises multiple modules, but the paper lacks the necessary ablation studies, making it difficult to assess the contributions of the individual components.
- Although the paper emphasizes robustness to noise, the experiments do not present performance curves across varying noise levels.

---

> ### Author Rebuttal · Authors · 2026-03-30
>
> > **W1: Unclear novelty: the method combines existing components within prior frameworks.**
>
> **R1**: Thank you for your insightful comment. Our method is not a direct combination of KFNN and LoMar, but targets a setting not addressed by prior work: density-based learning in FL under **privacy constraints, majority-client skew, and stochastic update noise**.
>
> KFNN relies on raw data for density estimation, violating FL privacy constraints. LoMar operates in parameter space but assumes fixed neighborhoods, which breaks under majority-client skew. Moreover, neither accounts for stochastic perturbations in client updates or provides stability guarantees.
>
> To address these gaps, we propose a unified framework with: (i) privacy-preserving density estimation in parameter space, (ii) adaptive neighborhood selection via density margins to handle skew, and (iii) Kalman-based filtering to stabilize estimation under noise, with theoretical support.
>
> These components are tightly coupled, removing any leads to consistent performance degradation, as evidenced by Table 1, 2 and Fig. 3, 5. To our knowledge, no prior work jointly addresses these challenges.
>
> ---
>
> > **W2: Unrealistic assumption of known client data sources in privacy-preserving FL.**
>
> **R2**: Thank you for the comment. Our method is fully **source-agnostic** and does not require or infer client data sources; “source” is only a conceptual abstraction for heterogeneity.
>
> All computations rely solely on model updates: density is estimated in parameter space, neighborhoods are defined by update similarity, and aggregation weights are derived from continuous density margins, without labels or side information.
>
> While cluster-like structures may emerge, they do not correspond to true data sources and introduce no additional privacy risk beyond standard FL. The framework operates without grouping in cross-device settings, and any grouping in cross-silo settings is optional and for interpretation only.
>
> ---
>
> > **W3: Limited comparisons with outdated baselines.**
>
> **R3**: Thank you for the suggestion. We agree that including more recent baselines would strengthen the evaluation and will add them in the revision.
>
> Our selection is guided by component-wise validation: FedAvg [2017] as the standard baseline; LoMar [2021] for parameter-space aggregation with fixed neighborhoods, validating our adaptive design; and FedLC [2022] and FedRDN [2025] to cover label and feature skew, demonstrating unified handling of heterogeneity.
>
> While recent methods often target specific aspects of heterogeneity, we will include additional up-to-date baselines to provide a more comprehensive comparison in the revised version.
>
> ---
>
> > **W4: Lack of Ablation Studies for Individual Components.**
>
> **R4**: Thank you for highlighting the importance of ablation analysis. We agree that full ablations are necessary. The current version provides partial evidence: varying neighborhood sizes (Table 1, Fig. 3) supports adaptive selection, and reduced margin variance (Fig. 5) validates Kalman filtering.
>
> To address this, we conduct end-to-end ablations on 4 datasets, comparing (i) w/o adaptive neighborhood selection, (ii) w/o Kalman filtering, and (iii) the full model. The full model consistently achieves the best performance across all datasets. Removing adaptive selection leads to notable drops (e.g., 0.739 to 0.630 on audiology), while removing Kalman filtering also degrades performance (e.g., 0.781 to 0.756 on autos), demonstrating complementary and non-redundant gains.
>
> | | anneal | audiology | autos | balance-scale |
> |:--------|:------:|:---------:|:-----:|:--------------:|
> | (i)     | 0.927  | 0.630     | 0.634 | 0.864          |
> | (ii)    | 0.957  | 0.717     | 0.756 | 0.880          |
> | (iii)   | 0.961  | 0.739     | 0.781 | 0.888          |
>
> We will include these results in the final version.
>
> ---
>
> > **W5: Missing Evaluation Across Varying Noise Levels.**
>
> **R5**: Thank you for highlighting this. To address this concern, we conduct additional experiments by varying the noise variance $\beta$ across four datasets. We define an improvement ratio $\Delta = \frac{|m - \tilde{m}| - |m - \hat{m}|}{|m - \tilde{m}|}$, where $m$, $\tilde{m}$, and $\hat{m}$ denote clean, noisy, and filtered margins. Larger $\Delta$ indicates greater error reduction. Results are shown as follows.
>
> | $\beta$     | mnist  | fashion-mnist | femnist | cifar-10 |
> |:------|:------:|:-------------:|:-------:|:--------:|
> | 0.01  | -0.039 | 0.294         | 0.280 | 0.065    |
> | 0.02  | 0.019  | 0.370         | 0.239   | 0.204    |
> | 0.05  | 0.127  | 0.457         | 0.251   | 0.361    |
> | 0.10  | 0.227 | 0.489   | 0.267   | 0.512 |
>
> The results show consistent improvements across noise levels, with controlled degradation as $\beta$ increases. Even when negative at very low noise (e.g., MNIST at $\beta=0.01$), it quickly becomes positive and improves, indicating strong robustness. We will include these results in the final version.

---

> > ### Author Rebuttal · Reviewer_rhsK · 2026-04-03
> >
> > Thank you for the detailed rebuttal and the additional experimental results regarding ablation and noise-level evaluation.
> >
> > While the rebuttal clarifies the target setting, the technical contribution still appears to be a combination of existing ideas within the FL framework. The rebuttal states that the method is fully source-agnostic, but this seems inconsistent with the current manuscript, where source-level structure is explicitly used in the formulation. The rebuttal does not substantially address my concern about limited comparison with more recent related methods. Therefore, based on the current version, I maintain my original score.

---

> > > ### Author Response · Authors · 2026-04-05
> > >
> > > >**1. Clarification on the combination of existing components concern.**
> > >
> > > **R1**: Thank you for this thoughtful comment. We respectfully disagree that FedVeer is a simple combination of existing ideas. This view overlooks the structural coupling inherent in the problem setting. Privacy-preserving federated learning with severe majority-client skew and stochastic update noise imposes interdependent constraints. Handling skew without stabilizing noise leads to unreliable margins, while noise handling without adaptive neighborhoods fails under skew. These challenges cannot be addressed independently. FedVeer is derived from this coupled structure rather than assembled from separate modules.
> > >
> > > **Conceptually**, FedVeer reformulates density estimation in parameter space, enabling learning from model updates instead of raw data. This shift in the estimation object is driven by privacy constraints and differs fundamentally from data-level or fixed-neighborhood methods.
> > >
> > > **Algorithmically**, adaptive neighborhood selection and Kalman-based stabilization are intrinsically linked. Max-margin learning determines neighborhood size, yet margin estimation is unstable under noisy updates without filtering. Filtering alone is ineffective without margin-driven adaptation. Together they form a unified optimization-stabilization mechanism.
> > >
> > > **Theoretically**, we prove uniqueness of the max-margin solution and derive a high-probability bound on noise-induced margin deviation, which is strictly tightened after filtering. The integration is thus theoretically grounded rather than heuristic.
> > >
> > > **Empirically**, removing either component consistently degrades performance. FedVeer improves accuracy by up to 6.36 percent and reduces noise-induced degradation by 6.01 percent, confirming that robustness stems from joint design.
> > >
> > > In sum, FedVeer is a tightly coupled, problem-driven framework rather than a juxtaposition of existing techniques.
> > >
> > > ---
> > >
> > > >**2. Clarification on the source-agnostic claim and source-level modeling.**
> > >
> > > **R2**: Thank you for the insightful comment. There is no contradiction between source-agnostic design and source-level modeling. The concern arises from treating "source" as a predefined or labeled entity, which our framework does not assume.
> > >
> > > In FedVeer, a source is an abstraction for distributional heterogeneity [1], grouping clients with similar update patterns in parameter space. Sources are inferred solely from observable signals such as model updates or parameter similarity. They may align with institutions in cross-silo settings (as illustrated in Fig. 1) or be inferred via clustering in cross-device settings. They are latent, data-driven, and require no ground-truth identities.
> > >
> > > Boundary clients may occur, but their inconsistent updates yield lower density and are naturally down-weighted. Thus, the method does not rely on precise source assignments. Source modeling formalizes structured heterogeneity [1] and is orthogonal to our contribution on mitigating majority-client skew under noise; neither the algorithm nor the theory assumes known source identities.
> > >
> > > We will clarify this in the revision.
> > >
> > > [1] Vo, T. V., et al. An Adaptive Kernel Approach to Federated Learning of Heterogeneous Causal Effects. In Advances in Neural Information Processing Systems 35 (NeurIPS 2022), New Orleans, Louisiana, USA, 2022. Curran Associates, Inc.
> > >
> > > ---
> > >
> > > >**3. Insufficient Comparison with Recent State-of-the-Art Methods.**
> > >
> > > **R3**: Thank you for the suggestion. We include two recent baselines: TPFL [2] (TDSC 2026), which relies on a dominance-based density assumption in gradient space, and FedLSA [3] (AAAI 2025), a representation-learning approach that addresses domain skew using local embeddings. Their results are shown below:
> > > |        | MNIST | Fashion-MNIST | FEMNIST | CIFAR-10 |
> > > |:-:|:-:|:-:|:-:|:-:|
> > > | FedLSA | 0.808 | 0.505 | 0.533 | 0.578 |
> > > | TPFL   | 0.833 | 0.678 | 0.672 | 0.581 |
> > > | FedVeer| 0.916 | 0.704 | 0.781 | 0.580 |
> > >
> > > FedVeer outperforms both methods on three datasets under majority-client skew, where dominance-based density assumptions may fail and feature-level alignment alone is insufficient for joint label skew and noisy updates. On CIFAR-10, although final accuracy is comparable to TPFL, FedVeer converges substantially faster. Convergence curves will be included in the final version.
> > >
> > > Overall, the gains stem from addressing a regime where dominance-based density assumptions break, rather than incremental refinement of prior methods.
> > >
> > > [2] Gu, K., et al. Two-dimensional privacy-preserving federated learning scheme against poisoning attacks. IEEE Transactions on Dependable and Secure Computing (TDSC), 2026.
> > >
> > > [3] Fu, L., et al. Beyond federated prototype learning: Learnable semantic anchors with hyperspherical contrast for domain-skewed data. In Proceedings of the AAAI Conference on Artificial Intelligence (AAAI 2025), Philadelphia, PA, USA, 2025. AAAI Press.

---

### Decision · Program_Chairs · 2026-04-30

**Decision:**

Accept (regular)

**Comment:**

This paper focuses on the challenge of data skew by proposing a skew-aware FL framework that utilizes self-adaptive kernel density estimation with k-free neighborhoods. The method is novel and theoretically sound. Although some reviewers may be concerned about the motivation of the designed mechanism, the response of authors also gives reasonable explanation.